# Projected health impact of post-discharge malaria chemoprevention among children with severe malarial anaemia in Africa

Lucy C. Okell [1] ✉, Titus K. Kwambai[2,3], Aggrey Dhabangi[4], Carole Khairallah[3], Thandile Nkosi-Gondwe[5,6], Peter Winskill [1], Robert Opoka[4], Andria Mousa [1], Melf-Jakob Kühl[7], Tim C. D. Lucas [8], Joseph D. Challenger [1], Richard Idro[4], Daniel J. Weiss[9,10], Matthew Cairns [11], Feiko O. ter Kuile [2,3], Kamija Phiri[5,6], Bjarne Robberstad[7] & Amani Thomas Mori [7,12,13] ✉

Children recovering from severe malarial anaemia (SMA) remain at high risk of readmission and death after discharge from hospital. However, a recent trial found that post-discharge malaria chemoprevention (PDMC) with dihydroartemisinin-piperaquine reduces this risk. We developed a mathematical model describing the daily incidence of uncomplicated and severe malaria requiring readmission among 0–5-year old children after hospitalised SMA. We fitted the model to a multicentre clinical PDMC trial using Bayesian methods and modelled the potential impact of PDMC across malaria-endemic African countries. In the 20 highest-burden countries, we estimate that only 2–5 children need to be given PDMC to prevent one hospitalised malaria episode, and less than 100 to prevent one death. If all hospitalised SMA cases access PDMC in moderate-to-high transmission areas, 38,600 (range 16,900–88,400) malaria-associated readmissions could be prevented annually, depending on access to hospital care. We estimate that recurrent SMA post-discharge constitutes 19% of all SMA episodes in moderate-to-high transmission settings.

Severe malarial anaemia (SMA) still contributes substantially to childhood mortality and morbidity in malaria-endemic countries in Africa. *P. falciparum* malaria causes anaemia by triggering severe haemolysis of erythrocytes and suppression of erythropoiesis[1]. In highly endemic areas, around one-third of all hospitalised children may be severely anaemic (Hb<5 g/dL)[2,3] and severe anaemia may contribute to around half of all malaria-attributed deaths[4]. While malaria transmission has declined in many countries in the past two decades, highly endemic conditions persist in parts of sub-Saharan Africa. 70% of global malaria deaths occurred in just ten countries in Africa in 2020[5]. These countries are now the focus of the WHO High Burden to High Impact (HBHI) initiative, aimed at accelerating progress in malaria control in the hardest-hit areas.

[1]MRC Centre for Global Infectious Disease Analysis, Department of Infectious Disease Epidemiology, Imperial College, London W2 1PG, UK. [2]Centre for Global Health Research (CGHR), Kenya Medical Research Institute (KEMRI), Kisumu, Kenya. [3]Department of Clinical Sciences, Liverpool School of Tropical Medicine (LSTM), Liverpool, UK. [4]College of Health Sciences, Makerere University, Kampala, Uganda. [5]Kamuzu University of Health Sciences, Blantyre, Malawi. [6]Training and Research Unit of Excellence, Blantyre, Malawi. [7]Section for Ethics and Health Economics, Department of Global Public Health and Primary Care, University of Bergen, P.O. Box 7804, 5020 Bergen, Norway. [8]Big Data Institute, University of Oxford, Oxford, UK. [9]Malaria Atlas Project, Telethon Kids Institute, Perth Children's Hospital, 15 Hospital Avenue, Nedlands, Australia. [10]Curtin University, Bentley, Australia. [11]International Statistics and Epidemiology Group, London School of Hygiene and Tropical Medicine, London, UK. [12]Chr. Michelsen Institute, P.O. Box 6033, N-5892 Bergen, Norway. [13]Muhimbili University of Health and Allied Sciences, P.O.Box 65001 Dar es Salaam, Tanzania. ✉e-mail: l.okell@imperial.ac.uk; pax_amani@yahoo.com

Between 0.4 and 13% of children with severe anaemia in low and lower-middle-income countries die during the acute hospitalisation phase, but those who survive also remain at high risk of readmission and death following hospital discharge despite having received appropriate care[6]. A recent meta-analysis found that the odds of death among previously severely anaemic children during the first six months after discharge is 72% higher than during hospitalisation and over two times higher than for those admitted with other conditions without severe anaemia[7]. Full haematological recovery following treatment of severe anaemia is thought to take around six weeks, and any malaria infections during this period substantially increase the risk of recurrent severe anaemia or death[8,9]. However, no specific interventions have been widely implemented to tackle this large post-discharge burden of morbidity and mortality among SMA patients.

Post-discharge malaria chemoprevention (PDMC) is the provision of full treatment courses of long-acting antimalarials administered at pre-determined time intervals after discharge from hospital among children recently admitted with severe anaemia. Two randomised clinical trials have shown that PDMC can reduce the high post-discharge malaria morbidity and mortality. In Malawi, three months of PDMC with monthly artemether-lumefantrine achieved a protective efficacy of 41% against the risk of death or all-cause readmission and 49% against clinical malaria in the first 3 months post-discharge[10]. More recently, a multicentre trial in moderate-to-high transmission areas of Uganda and Kenya showed that three months of PDMC with the longer-acting drug dihydroartemisinin-piperaquine (DP) reduces the risk of deaths or all-cause readmissions by 70% and hospitalised malaria episodes by 87% during the same period[11].

In June 2022, the World Health Organisation adopted PDMC in the malaria guidelines, recommending its use in areas with parasite prevalence >10%[12]. There is now a need to understand the potential epidemiological impact of implementing PDMC, to support economic evaluation of the intervention, and to plan how PDMC could contribute to initiatives such as the HBHI programme. In this study, we first develop a deterministic model to characterise the natural history of post-discharge malarial disease according to transmission intensity. We incorporate drug protection from PDMC into the model based on previous pharmacodynamic analysis and parameterise the model by fitting to data from a multicentre trial of PDMC. Next, we embed the post-discharge disease model within a population-wide model of severe malarial illness to understand the contribution of post-discharge malaria morbidity to the total disease burden, the potential demand for PDMC and its public health impact in malaria-endemic countries of Africa, taking into account different transmission intensities and access to healthcare.

## Results

### Malaria outcomes and impact of PDMC during 6 months post-discharge

Our analysis was informed by data from a previous trial of PDMC conducted between 2016–2018 in nine hospitals in areas with moderate-to-intense perennial malaria transmission in Kenya and Uganda, described in detail elsewhere[11]. In brief, 1049 children under five years of age who had been hospitalised for severe anaemia were randomised to receive either a three-day course of dihydroartemisinin-piperaquine (DP) ($N = 524$) or placebo ($N = 525$) at weeks 2, 6, and 10 after discharge from the hospital (i.e. 3 × 3 full DP courses = 9 DP doses for each child in total). All children had received the standard in-hospital treatment for severe anaemia including blood transfusion. 85% had malaria parasites and had been given parenteral artesunate. They then received a 3-day course of artemether-lumefantrine (AL) at the time of discharge. Multiple health outcomes were assessed over the next 6 months. PDMC caused a reduction in malaria outcomes, but not readmissions and out-patient hospital visits for other conditions. We therefore focus our analysis on incidence of malaria post-discharge, which was measured by passive follow-up in study clinics. Malaria cases were hospitalised again during follow-up if they required parenteral treatment, or if they had severe anaemia.

We developed a cohort model describing the daily incidence of uncomplicated and hospitalised malaria in the children after initial hospital discharge. We allowed incidence to depend on the entomological inoculation rate (EIR; infectious bites per person per year), the time since discharge, the rate of antimalarial treatment for symptomatic malaria episodes (allowing for post-treatment prophylaxis), and receipt of DP for PDMC. DP prophylaxis is modelled as a probability of prevention of reinfection that declines over time since treatment, dropping to 50% protection after 26 days (Fig. S1)[13]. We also allowed that if children develop symptomatic malaria during DP prophylaxis, the drug might reduce the severity of illness (as indicated by the probability of hospitalisation). All PDMC doses in the trial except the first were administered at home and we allowed for imperfect adherence (Table S1). A total of 333 hospitalised and 557 uncomplicated malaria episodes were recorded during the 6 months post-discharge. The model was fitted to individual patient records using Bayesian methods, allowing for loss to follow-up rates observed in the data. We used semi-informative prior values for the EIR in each hospital site based on Malaria Atlas Project maps[14].

The model was able to accurately capture the total number of observed uncomplicated and hospitalised malaria events in both the post-discharge cohort placebo and PDMC trial arms (Fig. 1). The modelled protective efficacy of PDMC was 86.7% against hospitalised malaria and 73.6% against uncomplicated malaria during 3–14 weeks post-discharge. This was close to the trial estimates of 87% and 69% for hospitalised and uncomplicated malaria, respectively. The model fit the data better when it was assumed that PDMC reduces the severity of any malaria illness even if infection occurs, such that the percentage of symptomatic malaria cases that required hospitalisation was 24% (95% credible interval (CI) 16–33%) among children who had taken PDMC within the past 40 days, compared to 38% (95% CI 35–42%; Table S1) among children with no recent PDMC. The model was also able to capture the change in daily incidence rates over time since discharge from hospital (Fig. 1B, C). The risk of uncomplicated and hospitalised malaria per infectious bite in the placebo group declined by approximately 50% between the beginning and the end of follow-up (Fig. 1A and S4). The incidence of uncomplicated and hospitalised malaria increased with EIR, but the average risk of these outcomes per infectious bite decreased (Fig. 2, S5), similar to relationships previously observed in the general population[15,16]. Three previous post-discharge studies which were not used during model fitting[10,17,18] showed a similar or slightly lower incidence of post-discharge hospitalised malaria than predicted by our model, given the estimated local EIR (Fig. 2B).

The incidence of hospitalised malaria was strikingly high in the post-discharge placebo group compared to the average in the general population of the same age estimated in other studies[19,20] (Fig. 2B), being 18–60 times higher during post-discharge weeks 3–14 in settings with parasite prevalence>10%, and still 10–36 fold higher in weeks 15–25. This high incidence suggests persistent vulnerability of these children beyond the end of the PDMC intervention at week 14. The incidence of uncomplicated malaria was 1.2–2.5 times higher than expected in the general population of the same age in weeks 3–14. The total incidence of symptomatic malaria episodes (both uncomplicated and hospitalised) was more than the expected incidence of infectious bites in 0–5-year old in four trial sites, suggestive of higher than average exposure to mosquitoes (e.g. due to spatial heterogeneity in transmission, lack of protective measures such as bed nets, etc) (Fig. S6).

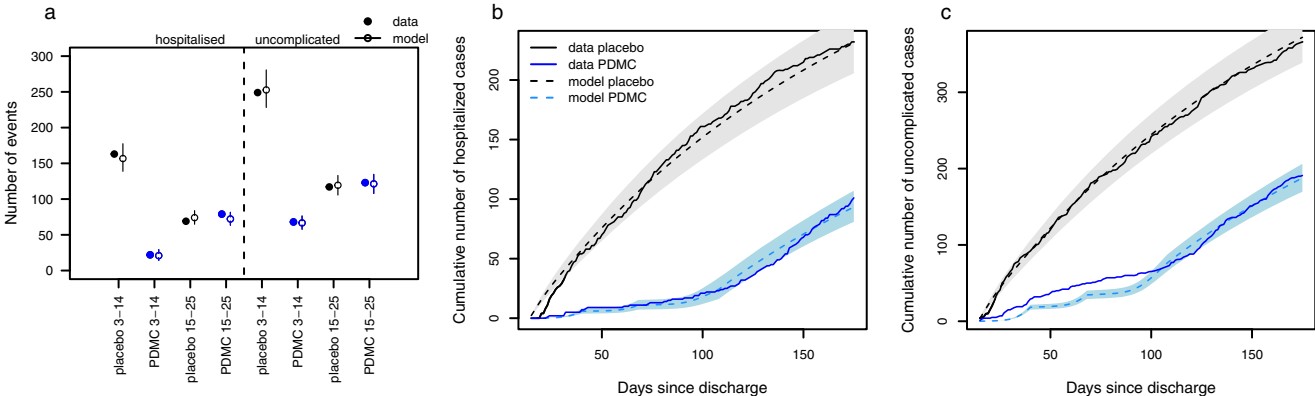

**Fig. 1 | Model fits to trial data on number of malaria cases across all sites.**
**a** Hospitalised (left) and uncomplicated malaria (right) in weeks 3–14 and 15–25 post-discharge in PDMC (blue) and placebo (black) trial arms. 333 hospitalised and 557 uncomplicated malaria episodes were experienced in total among 524 children in the PDMC group and 525 children in the placebo group. Model predictions show medians and 95% credible intervals (CI) (see Table S4 for data and model predictions). **b** Cumulative daily number of hospitalised malaria cases by time since discharge after the original SMA episode in placebo (black) and PDMC (blue) trial arms; solid line=data, dashed line and shaded area=model median fit and 95% CI. (**c**) As (**b**), for uncomplicated malaria cases.

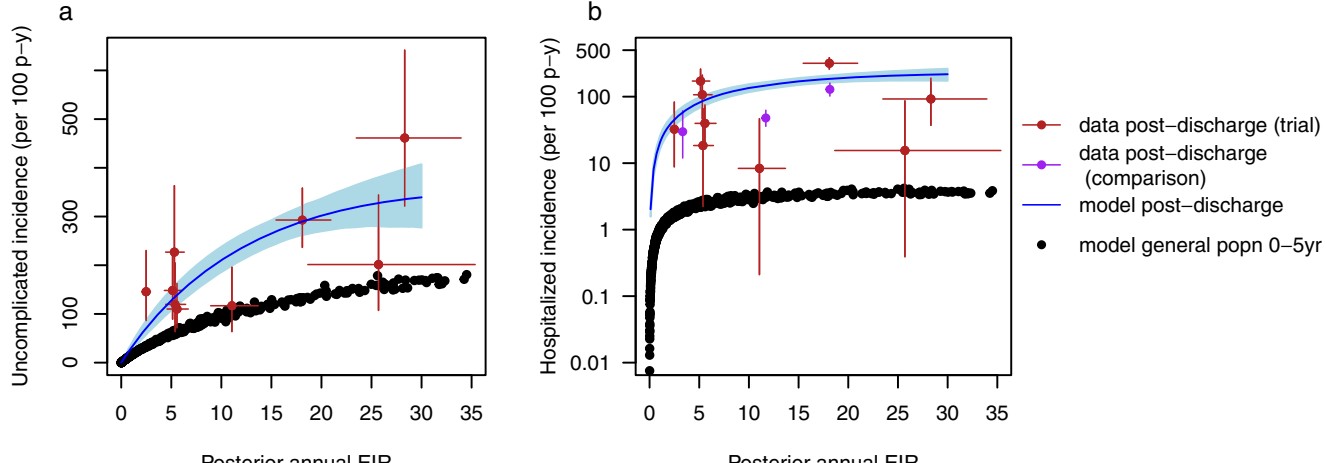

**Fig. 2 | Model fits to trial data by site. a** Relationship between posterior median estimated EIR values and incidence of uncomplicated malaria per 100 person years in 0–5-year-olds 3–14 weeks after hospital discharge following severe malarial anaemia. The data are from the 9 trial locations in Uganda and Kenya (red; data post-discharge (trial) = data from Kwambai et al.; N = 249 cases, Table S5)[11]. Vertical error bars = 95% confidence interval of the incidence data; horizontal error bars = 95% credible interval of posterior EIR estimate). The median model prediction is in blue (shaded area = 95% credible interval). For comparison, the estimated incidence of uncomplicated malaria by EIR in the general population of under five-year-olds is shown in black. **b** As (**a**) showing hospitalised malaria episodes in the same trial group (log scale; N = 163 cases, Table S5). For comparison, the estimated incidence of hospitalised malaria in the general population of children aged 0–5 years for a given annual EIR is shown in black. Three additional validation data points are shown from separate post-discharge studies (purple; Uganda: Opoka et al.[17,18] and Malawi: Phiri et al.[10]) which were not used for fitting the model. Posterior predictive checks of the model against the data in each site are shown in Fig. S8.

## Impact of PDMC across sub-Saharan Africa and burden of recurrent SMA

To predict PDMC demand and impact in different settings, we extended the modelling framework to describe severe malarial anaemia (SMA) in the total general population of under-five-year old and embedded our post-discharge cohort model within this (Methods and Supplementary Appendix). In the model, when an individual experiences an episode of SMA, they then enter a 6-month period with high risk of recurrent malaria, as estimated from the trial. We allowed for SMA cases in the community who do not access treatment in hospital, as well as lower treatment coverage for uncomplicated malaria and lower adherence to PDMC under routine healthcare outside the trial setting. As a model input, we generated estimates of the total incidence of hospitalised SMA in 0–5 year olds for each subnational region (first administrative unit) of endemic countries in Africa based on a recently published analysis correlating hospital SMA cases with local malaria infection prevalence (Fig. S2)[21]. We used Malaria Atlas Project estimates of infection prevalence for each region in 2019[14]. We also generated estimated annual EIR and incidence of total hospitalised malaria (SMA and other types) for each region using the Imperial College malaria transmission model[22]. We used the model to calculate the proportion of all hospitalised SMA episodes which are recurrent (within 6 months of a previous SMA episode), and the potential impact of PDMC upon these.

We predict the impact of PDMC will be greatest in countries with higher transmission intensities (Fig. 3A, Table 1). In areas with parasite prevalence in 2–10-year old (PfPR$_{2-10}$) >10%, where PDMC is recommended, we estimate that a median of 0.21 (range 0.04–0.47) malaria-associated readmissions could be prevented per child given the 3 courses of PDMC, weighting for population size in each area (Fig. 3C). This means that a median of 4.8 children need to be given PDMC to prevent one hospitalised malaria episode (Fig. 3B). In the two highest burden countries, Nigeria

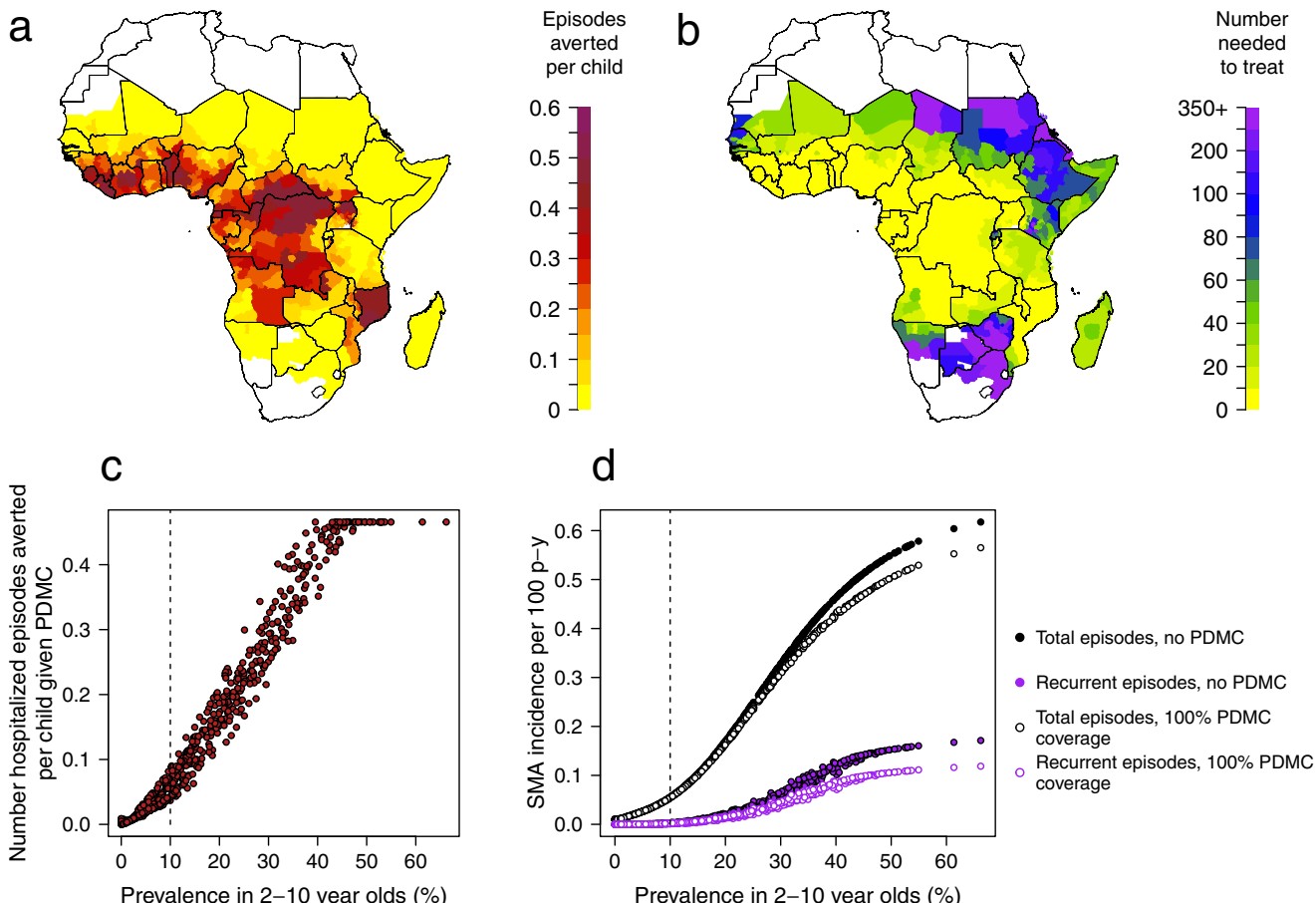

**Fig. 3 | Impact of PDMC. a** Average number of hospitalised malaria episodes averted during the 6 months post-discharge period per child aged 0–5 years given PDMC. **b** Number needed to treat with PDMC to avert 1 malaria episode requiring hospitalisation. All estimates are shown for subnational (administrative area 1) regions and incorporate imperfect adherence to the three prescribed courses of PDMC as observed in ref. [23]. The assumption in these results is that 50% of cases requiring hospitalisation access hospital care, but there is negligible change in these outputs when this percentage is varied from 30–70%. **c** Parasite prevalence in 2–10-year old versus average number of hospitalised malaria episodes averted during the 6 months post-discharge period per child aged 0–5 years given PDMC (results from **a**). Dashed line indicates parasite prevalence above which WHO recommends PDMC[12]. **d** Total and recurrent SMA episodes per 100 person years with and without PDMC (including cases who do not access hospital care as well as those who do). Model estimates are shown in the absence (solid circles) and presence (open circles) of PDMC. Recurrent episodes are those occurring within 6 months of a previous SMA episode. PDMC is given 100% coverage to hospitalised cases only and impact largely occurs in the first 3 months post-discharge. We assumed that 50% of individuals with SMA reach hospital and varied this for sensitivity analysis (Fig. S9). Assuming a lower proportion hospitalised results in larger total estimates of SMA burden in the absence of PDMC, and smaller total impact of PDMC.

and the Democratic Republic of the Congo (DRC)[5], only 3.1 and 2.9 children need to be given PDMC to prevent a hospitalised episode, respectively (Table 1). These results assume the lower adherence to PDMC as observed during an implementation study (77–89% depending on dose; Table S2) compared to the near-perfect adherence observed in clinical trials[23]. This lower adherence is estimated to reduce protective efficacy against hospitalised malaria during weeks 3–14 from 87 to 76%, resulting in an overall protective efficacy over the 6 months of 53% (as opposed to 62% with perfect adherence). The relatively high overall efficacy over the 6-month period results from PDMC taking effect in the earlier post-discharge period when malaria risk is highest.

Predictions of total malaria readmissions and deaths averted by PDMC are sensitive to assumptions about the proportion of malaria cases who access hospital care when required (since PDMC is only provided to hospitalised cases), and the case fatality rate outside hospital settings. The probability of malaria patients accessing hospital care is challenging to estimate and varies between settings. The following results assume a base scenario where 50% of malaria cases requiring hospitalisation actually access it, and a range is obtained

from assuming 30–70% access based on community studies of severe disease[24]. For hospitalised SMA and non-SMA malaria episodes, case fatality rates of 7.4% and 1.0% were assumed, respectively[25–27]. The case fatality among the non-SMA hospitalised cases is lower because we allow that not all admitted patients meet the strict WHO criteria for severe malaria[27], which would have a higher case fatality rate. We assumed the case fatality rate of malaria cases unable to access hospital would be doubled based on a previous analysis by Camponovo et al.[27].

If all hospitalised children aged 0–5 years with SMA were given PDMC in areas with $PfPR_{2-10} > 10\%$, we estimate that a total of 38,600 (range 16,900–88,400) malaria-associated readmissions could be prevented per year across all modelled malaria-endemic countries, and 2176 deaths prevented per year (range 1078–4315). Three-quarters of these prevented readmissions and deaths are in the ten countries that are the focus of the WHO High Burden High Impact programme[5]. In areas with $PfPR_{2-10} > 10\%$, we estimate that a mean of 112 children need to be given PDMC to prevent one death (range 99–130). In Nigeria and the DRC, 55 (range 48–64) and 53 (range 47–562) children need to be given PDMC to prevent one death, respectively. Including local

**Table 1 | Estimated PDMC impact and demand in sub-Saharan African countries**

| Country | Population aged under five years in areas of PfPR$_{2-10}$ > 10% | Incidence of hospitalised severe malarial anaemia per 100 person years in 0-5-year-olds without PDMC (range[a]) | Average number of hospitalised malaria episodes prevented per 100 children given PDMC | Number of children needed to treat with PDMC to prevent one hospitalised malaria episode[b] | Number of children needed to treat with PDMC to prevent one malaria death (range[a])[b] | Annual PDMC demand (range[a]) |
|---|---|---|---|---|---|---|
| HIGH BURDEN TO HIGH IMPACT COUNTRIES | | | | | | |
| Burkina Faso | 3,011,302 | 0.16 (0.11–0.26) | 24.1 | 4.2 | 74 (65–85) | 2277 (971–5341) |
| Cameroon | 3,694,640 | 0.25 (0.18–0.41) | 19.3 | 5.2 | 92 (81–106) | 4398 (1877–10,306) |
| Democratic Republic of the Congo | 20,216,757 | 0.32 (0.23–0.52) | 33.3 | 3.0 | 53 (47–62) | 31,884 (13,568–74,923) |
| Ghana | 4,362,206 | 0.18 (0.13–0.3) | 21.0 | 4.8 | 85 (74–98) | 3845 (1640–9014) |
| Mali | 2,548,410 | 0.1 (0.07–0.17) | 10.7 | 9.3 | 166 (146–191) | 1235 (528–2889) |
| Mozambique | 4,149,401 | 0.31 (0.22–0.5) | 38.5 | 2.6 | 46 (41–53) | 6325 (2689–14,877) |
| Niger | 3,238,326 | 0.14 (0.1–0.23) | 18.9 | 5.3 | 94 (82–109) | 2147 (916–5032) |
| Nigeria | 31,310,407 | 0.27 (0.2–0.44) | 32.7 | 3.1 | 54 (48–63) | 41,445 (17,638–97,377) |
| Tanzania | 2,145,631 | 0.07 (0.05–0.12) | 11.8 | 8.5 | 151 (133–174) | 736 (315–1722) |
| Uganda | 6,293,959 | 0.22 (0.16–0.37) | 32.4 | 3.1 | 55 (48–63) | 6962 (2963–16,356) |
| OTHER MALARIA-ENDEMIC COUNTRIES | | | | | | |
| Angola | 4,577,253 | 0.16 (0.12–0.27) | 20.2 | 4.9 | 88 (77–101) | 3632 (1549–8512) |
| Benin | 2,089,789 | 0.38 (0.28–0.63) | 37.2 | 2.7 | 48 (42–55) | 3984 (1694–9368) |
| Burundi | 1,290,817 | 0.15 (0.11–0.24) | 18.6 | 5.4 | 96 (84–110) | 904 (386–2118) |
| Central African Republic | 942,760 | 0.36 (0.27–0.59) | 39.5 | 2.5 | 45 (40–52) | 1714 (728–4032) |
| Chad | 1,692,264 | 0.08 (0.06–0.13) | 6.6 | 15.2 | 271 (239–312) | 621 (266–1452) |
| Cote d'Ivoire | 4,024,794 | 0.34 (0.25–0.56) | 33.9 | 3 | 52 (46–61) | 6789 (2889–15,956) |
| Equatorial Guinea | 133,034 | 0.44 (0.32–0.72) | 43.1 | 2.3 | 41 (36–48) | 296 (126–697) |
| Gabon | 387,369 | 0.26 (0.19–0.42) | 24.2 | 4.1 | 73 (65–85) | 482 (206–1131) |
| Guinea | 1,905,031 | 0.21 (0.15–0.35) | 23.7 | 4.2 | 75 (66–87) | 1977 (843–4636) |
| Guinea-Bissau | 29,867 | 0.07 (0.05–0.11) | 8 | 12.6 | 223 (203–258) | 9 (4–22) |
| Kenya | 1,954,633 | 0.17 (0.13–0.29) | 26.4 | 3.8 | 67 (59–78) | 1659 (707–3893) |
| Liberia | 642,712 | 0.38 (0.28–0.63) | 40.9 | 2.4 | 43 (38–50) | 1236 (525–2909) |
| Malawi | 2,699,188 | 0.12 (0.08–0.19) | 18.1 | 5.5 | 98 (86–113) | 1524 (651–3571) |
| Mauritania | 364,450 | 0.09 (0.06–0.14) | 5.8 | 17.3 | 307 (274–355) | 147 (63–344) |
| Republic of Congo | 585,972 | 0.3 (0.22–0.49) | 32.2 | 3.1 | 55 (48–64) | 858 (365–2017) |
| Sierra Leone | 947,825 | 0.31 (0.23–0.51) | 39.2 | 2.6 | 45 (40–52) | 1476 (627–3472) |
| Somalia | 234,338 | 0.06 (0.04–0.1) | 4.4 | 22.6 | 401 (347–460) | 64 (27–150) |
| South Sudan | 2,437,409 | 0.23 (0.17–0.38) | 23.9 | 4.2 | 74 (65–86) | 2704 (1153–6341) |
| Togo | 1,262,166 | 0.22 (0.16–0.36) | 22.7 | 4.4 | 78 (69–90) | 1352 (576–3169) |
| Zambia | 1,920,552 | 0.11 (0.08–0.18) | 23 | 4.4 | 77 (68–90) | 1036 (442–2429) |
| **TOTAL** | | | | | | **133,719 (56,932–314,058)** |

[a]Range obtained from varying the proportion of cases hospitalised from 30–70%.
[b]Children are provided with full PDMC (3 courses of DP, each containing 3 doses) at discharge, but estimates allow for imperfect adherence in routine settings.
The ten high-burden countries targeted by the WHO High Burden to High Impact strategy are shown first. We include only subnational regions with parasite prevalence in 2–10 year olds (PfPR$_{2-10}$) > 10% according to WHO recommendations on PDMC, and weight by population size to obtain overall numbers for each country. We assume that demand is not affected by adherence, i.e. that all doses are given out although not necessarily taken. Estimates of demand are thus shown for 100% PDMC coverage (receipt of drugs), while for impact we adopted the adherence actually observed by Nkosi-Gondwe and colleagues during their implementation study[23].

seasonal variation in malaria incidence caused negligible change to the overall results, due to the long term nature of PDMC chemoprevention and the time lag between first SMA episode and chemoprevention (Fig. S11 and Table S3).

We estimate that in the absence of PDMC the burden of recurrent SMA episodes within 6 months of the original episode is a mean of 19% (range 3–28%) of the total SMA cases in <5-year-old children in areas with PfPR$_{2-10}$ > 10%. PDMC prevents an estimated 5.8% of all SMA in under fives when PfPR$_{2-10}$ > 10% in the base scenario where 50% are hospitalised, ranging from 3.5–7.9% under different access to hospital (Fig. S9).

### PDMC demand forecast
We estimate there are 133,700 (range 56,900–314,100) children under 5 years old per year who are hospitalised with SMA and survive, across all subnational regions with PfPR$_{2-10}$ > 10% in Africa (Table 1; Fig. S10). The highest number of children eligible for PDMC would be in Nigeria with 41,400 per year (range 17,600–97,400). In total, the ten high-burden to high-impact countries in Africa identified by WHO[5] would require PDMC for 95,300 children per year (range 41,500–218,800) in areas with PfPR$_{2-10}$ > 10%. Some countries not on this top-10 list also have high PDMC requirements, such as Côte d'Ivoire (Table 1).

## Discussion

Our model captures the high burden of repeat malaria episodes requiring hospitalisation within the first 6 months post-discharge among children recovering from an initial episode of SMA. We estimate that this risk is approximately 18–60 times higher than the average for children of the same age in the general population in PDMC-eligible areas. These repeat episodes constitute an estimated 19% of all SMA episodes in these settings. PDMC with monthly dihydroartemisinin-piperaquine targets this high-risk group and effectively prevents malaria-associated hospital readmissions or death in trials[11]. We predict that this effect would be notable across most endemic settings in Africa with an effective routine implementation of the intervention. PDMC would have the largest impact in the highest burden settings; e.g. in Nigeria and the DRC, where we estimate only 3 children need to be given PDMC to prevent a hospitalised malaria episode. However, in the large majority of settings with $PfPR_{2-10} > 10\%$, fewer than 10 children need to be given PDMC to prevent one hospitalised malaria episode. This dynamic is partly driven by the 'self-targeting' nature of PDMC–that only high-risk children admitted to hospital will receive the treatment (unlike other forms of chemoprevention, which are administered to all healthy children in a particular age range). Our analysis also suggests PDMC will be highly cost-effective across a wide range of settings, given that the cost of inpatient malaria care in sub-Saharan Africa is $15.64–$137.87[28]. However, cost-effectiveness will depend on the local costs of clinical management and the organisation of healthcare services.

The risk of hospitalised malaria remained high over a prolonged period, with the incidence still being 10–36 times higher than the average for the general population of the same age in the period 4–6 months after the original SMA episode. Earlier studies suggested that haematological recovery after severe anaemia requires around six weeks[9]. However, our findings suggest that recently discharged children remain at higher risk than the general population well beyond this period. Since the PDMC intervention effect is largely restricted to the first 14 weeks, providing longer protection may have significant health impacts. Future studies should assess the benefit of augmenting PDMC with additional prevention strategies against malaria such as longer courses of chemoprevention, malaria vaccines, or monoclonal antibodies that provide at least 6 months of protection. Caregivers in the PDMC trials were encouraged to use insecticide-treated nets for their children, but the provision of a new effective insecticide-treated net might have higher impact. Older children with SMA also have notable post-discharge morbidity[7], therefore increasing the eligible age range might also increase total impact.

Our results on the number of cases averted per child given PDMC and the number needed to treat to avert uncomplicated and hospitalised episodes were derived from clinical trial results and are relatively robust to model assumptions. However, the total demand for PDMC and the total number of cases and deaths that could be prevented are uncertain, given that PDMC is a hospital-based intervention and there is a lack of data on the probability of accessing hospital care. Access to hospitals is likely to vary greatly between and within countries[27]. A recent study tracking children with suspected severe malaria in the community in Uganda, Nigeria, and DRC found that only 41–65% access hospital care even after referral by a community health worker[24]. There are similar findings in LMIC settings relating to other diseases; for example in Lusaka, Zambia, only around one-third of fatal respiratory syncytial virus cases in infants sought care in hospital before death[29]. In our analysis we varied the percentage who access hospital care from 30–70% to allow for this uncertainty; however, it is possible that in some countries, access is lower or higher, meaning that we over- or underestimate both demand and impact of PDMC. Care seeking for repeat episodes of malaria after hospital discharge during the trial was probably better than in routine clinical practice because participants were financially remunerated for their travel expenses and may have had greater awareness of malaria danger signs. Although PDMC will produce a lower total impact where many children with SMA do not access hospital care, we find that the benefit per child given PDMC is greater in areas where children are less likely to return to the hospital for subsequent episodes, due to the high burden of recurrent severe malaria and its high case fatality rate in the absence of hospital care. Preventing repeat episodes is also important given that households can incur catastrophic health expenditure for severe malaria[30].

Our results are also sensitive to the assumed level of adherence to PDMC during routine implementation, which we based on an implementation trial that simply provided all treatments at discharge without further reminders[23]. 77–89% adherence was observed (adherence varied across the 1st–3rd treatments). If lower adherence was achieved during routine implementation, we could over-estimate impact of PDMC. However, high adherence has generally been observed in other malaria chemoprevention interventions. For example routine seasonal malaria chemoprevention (SMC) programmes in the Sahel region report adherence of 87–99%[31]. Further investigations will be needed to understand the interaction between SMC and the impact of PDMC. Studies in Burkina Faso and Mali have found that in the context of SMC, hospitalised cases peaked either at the beginning of the transmission season (prior to SMC), or late in the season (after SMC), but that repeat hospitalised episodes were relatively rare in the presence of SMC protection and good access to treatment within a research trial[32].

Mortality impact of PDMC remains a topic to be studied further in future, given that PDMC trials to date have not been powered to evaluate impact on mortality. Overall, 3 months PDMC with DP was associated with a large drop in all-cause mortality in the trial of 94% during the 3-month intervention period, but over the whole 6 months follow up there was a non-significant 35% reduction in mortality (95% CI −38, 70%)[11]. Given that the vulnerable group targeted by PDMC is small, we estimate PDMC would prevent only a fraction of total population SMA cases (4–8% in high transmission areas) and a smaller fraction of all malaria deaths (WHO estimates ~481,600 malaria deaths in under fives in 2020)[5]. This highlights the importance of implementing PDMC alongside key preventive interventions across the whole community.

Our finding that around 19% of hospitalised SMA episodes would occur within 6 months of the original episode is plausible considering previous hospital-based studies of children with severe anaemia. In Uganda, 75% of children with severe anaemia had been hospitalised within the previous 6 months for SMA[33]. In the PDMC trial in Kenya and Uganda, 35% of children had been hospitalised previously for severe disease of any cause[11]. We found that children with SMA are probably exposed to infectious mosquito bites more frequently than the average child, because the incidence of symptomatic malaria post-discharge was higher than the expected incidence of infectious bites in this age group in some settings (Fig. S6). The probability of developing infection and symptoms per infectious bite is also higher than in other children. The group who develop SMA may have particular frailty to malaria for genetic or biological reasons[7,34]. or other reasons that future studies may attempt to investigate.

Our model was able to replicate the observed total number of hospitalised and uncomplicated malaria episodes in the post-discharge trials. We estimated a relationship between EIR and both outcomes, although there was considerable variation around this in the data. There is uncertainty in the EIR values, which are based on predicted prevalence estimates from a geospatial model[35]. Uncertainty in the incidence of post-discharge episodes also arises from small sample sizes in some trial hospitals, as well as some misclassification of uncomplicated malaria. One outlier was Kamuli mission hospital in Uganda, which was predicted to have a high EIR yet showed a relatively

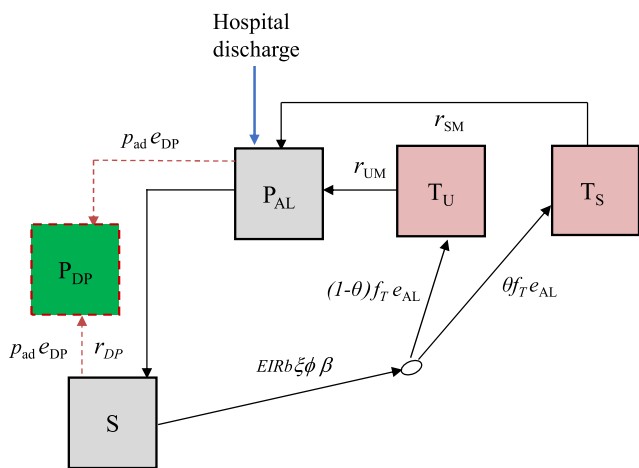

**Fig. 4 | Post-discharge cohort model.** Boxes represent health states, while arrows and labels show health state transition rates and probabilities of an event. Definitions are given in the text and Table S1. See supplementary information for further details.

low incidence of post-discharge hospitalised malaria. This hospital is a private not-for-profit facility, where admitted patients usually incur relatively higher out-of-pocket payments than patients at public hospitals. Trial personnel reported that some malaria cases that would otherwise have been hospitalised were treated as out-patient cases with injectable artesunate to avoid hospital costs and were therefore classified as uncomplicated malaria cases (personal communication, author A. Dhabangi). Some further misclassification could have occurred due to extended circulation of parasite antigens resulting in positive RDTs after successful treatment for a malaria episode. Although efforts were made during the trial to minimise this effect using blood smear confirmation of RDT-positive cases occurring within 2 weeks of a previous episode, another recent study showed the positive predictive value of an RDT relative to microscopy is still under 70% 5 weeks after an episode[32], meaning that the incidence of uncomplicated malaria could be slightly overestimated. This effect may be stronger in the PDMC arm given their additional intake of drugs as part of the intervention, which would result in a higher prevalence of recently cleared infection. This may explain the slight deviation of the model from the data in in the PDMC arm during the intervention period (Fig. 1).

Resistance to piperaquine has emerged in the Greater Mekong region, where DP was used for first-line therapy of uncomplicated malaria. There are concerns that the use of DP for chemoprevention could lead to the emergence of piperaquine resistance in Africa. However, the very small proportion of parasites that would be exposed to DP through PDMC programmes likely lowers the risk of resistance considerably. For example, in Nigeria, the estimated number of children eligible for PDMC each year, 41,400, is only ~0.1% of total malaria cases reported by the country to WHO[5]. Introduction of DP as a first-line treatment option in a number of African countries (Cameroon, Ghana and Nigeria[5]) is likely to substantially outweigh the use of DP for PDMC in terms of selecting for resistance.

In summary, our findings support the widespread implementation of PDMC in malaria-endemic countries in Africa, particularly high-burden countries. In countries with low burden, this consideration is likely to be influenced by intervention costs, which may be considered in economic evaluation. The total impact of PDMC is heavily dependent on the proportion of SMA cases that can access hospital care and the successful delivery of dihydroartemisinin-piperaquine. Further, the high risk of malaria morbidity persists for at least 6 months post-discharge, suggesting the need for longer-acting interventions.

## Methods

### Post-discharge cohort model

We initially considered using an existing well-established malaria transmission model to describe the incidence of malaria outcomes post-discharge[22]. However, preliminary analysis indicated that the high incidence of uncomplicated and hospitalised malaria after a severe malarial anaemia (SMA) episode, as observed in trials and clinical studies[11,17], was not adequately captured using this existing model, despite allowing for individual variation in immunity and exposure to mosquito bites. This is because the model was calibrated against population level incidence data, not against longitudinal data on risk within individuals over time. For the current analysis, a novel model was created to describe the natural history of malaria illness in the post-discharge population, how it changes over time and with varying malaria transmission intensities.

We developed a deterministic, discrete-time compartmental model with the following states: prophylaxis ($P_{DP}$ and $P_{AL}$), susceptible ($S$), treated uncomplicated malaria ($T_U$), and treated hospitalised malaria ($T_S$) (Fig. 4).

Children are given artemether-lumefantrine (AL) at the time of hospital discharge to clear any remaining malaria parasites, which provides a period of prophylaxis against reinfection. Therefore, children enter the model into the protected state $P_{AL}$ on the first day of their treatment. We modelled this initial period of AL protection as gamma-distributed with an average of 13 days based on previous analysis of reinfection during clinical trials of AL[36]. After prophylaxis, children in the placebo group enter the susceptible state $S$ and experience symptomatic malaria (uncomplicated or hospitalised) at a rate equal to the product of the local entomological inoculation rate (EIR), the probability that an infectious bite leads to infection $b$, the probability of symptoms $\phi$ and the relative exposure to bites among the post-discharge group of children ξ compared to the average child (Table S1). We only observe the incidence of symptomatic malaria, so these parameters are not individually identifiable and during model fitting, we estimate them as a product $b\phi\xi$. We assume that each infectious bite can only cause one symptomatic episode. Upon developing symptoms, we allowed a two-day delay for treatment seeking. Uncomplicated cases enter state $T_U$ and then go on to receive AL treatment which has 98% efficacy[5] and provides 13 days of prophylaxis (state $P_{AL}$). A proportion $\theta$ of symptomatic malaria cases experience disease sufficiently severe to be readmitted to hospital (they need parenteral antimalarials or have severe malarial anaemia), and enter a treated severe state $T_S$ which includes a mean hospital stay of 3 days based on the trial data[11]. As before, AL is then given at discharge, and the children enter state $P_{AL}$. Hospitalised cases include those who meet the strict WHO case definition of severe malaria as well as those who do not[27], in line with the outcome metric in the PDMC trial used for fitting the model[11]. We assumed that all uncomplicated and severe malaria episodes during follow-up were detected and treated. This is a reasonable assumption because the trial reimbursed participant costs of treatment seeking at study clinics, diminishing financial barriers to care. We do not track asymptomatic infections and those infected without symptoms remain in the $S$ state, which represents those susceptible to clinical malaria disease.

We further allow that the risk of symptomatic malaria per infectious bite could decline over time since hospital discharge, based on results from the PDMC trial, other post-discharge studies, and the hypothesis that recovery will reduce vulnerability[8,11]. We estimated the rate of the decline by allowing total incidence in the cohort to be scaled by a Weibull survival curve:

$$e^{\left[-\left(\frac{t_d}{\lambda_{\text{risk}}}\right)^{\eta_{\text{risk}}}\right]} \quad (1)$$

where $t_d$ is the time since hospital discharge and the scale and shape parameters $\lambda_{risk}$ and $\eta_{risk}$, are estimated.

The average probability that an infectious bite leads to successful infection and then to symptoms are both known to decline with increasing transmission intensity in the general population in any given setting due to acquired immunity and density-dependent effects, e.g. competition between parasites. We included this possibility by further scaling total incidence in each location by a similar functional form to that identified in previous analyses[15,37] of the relationship of EIR and the probability of symptomatic malaria:

$$e^{-w\text{EIR}} \tag{2}$$

and estimated the parameter $w$, since the relationship is unknown in this post-discharge population.

We track children in the PDMC trial arm separately. The model describing these children is the same as the placebo group except for additional PDMC protection against uncomplicated and hospitalised episodes. PDMC is given as three full courses of dihydroartemisinin-piperaquine (DP) starting at the beginning of weeks 2, 6 and 10 post-discharge. We modelled DP prophylaxis as a probability of prevention of reinfection that declines over time since treatment, using a Weibull-survival function as in previous work:[36]

$$e^{\left[-\left(\frac{t_{DP}}{\lambda_{DP}}\right)^{\eta_{DP}}\right]} \tag{3}$$

where $t_{DP}$ is the time since the last PDMC treatment, and $\lambda_{DP}$ and $\eta_{DP}$ represent the scale and shape parameters of the Weibull distribution. We allowed for different adherence to each of the three courses of PDMC based on trial data when fitting to the trial (Table 1, main text). For simplicity, we assumed that for each course of PDMC, caregivers either gave all three doses or none, which was relatively consistent with observations during implementation (only 1.5% of children received 1–2 doses of DP per treatment course, with the remainder taking all or none)[23]. The DP protection is applied to all children in the PDMC intervention group, except children in states $T_U$, $T_S$ and $P_{AL}$ (currently symptomatic and treated). When children leave the $T_U$ and $T_S$ states and return to $S$, we assume they have the same protection as those who did take PDMC. This approximately captures the trial recommendation that children should take PDMC once recovered from their symptomatic episode[38], although it was not possible to explicitly include multiple PDMC timings outside the standard trial times due to computational constraints. We further explored whether current PDMC drug protection could reduce the probability of needing hospitalisation, $\theta$. We defined 'current drug protection' as inhibitory drug levels providing >1% probability of protection.

## Model fitting and validation

We fitted the model to the trial data described above using Bayesian methods. Semi-informative priors for the EIR in each hospital catchment area were based on the Malaria Atlas Project (MAP) estimates of transmission intensity within 20 km of the hospital location (Table S1)[14]. We ran the model for each site and fitted to the daily incidence of uncomplicated and hospitalised malaria simultaneously across sites over the 6 months of follow-up. Since hospitalised malaria numbers were small in some sites, this method borrows information from the uncomplicated malaria cases to inform the posterior EIR. The parameters estimated during fitting were: the maximum incidence of symptomatic malaria (uncomplicated and hospitalised) per infectious bite at the beginning of post-discharge follow-up, the EIR in each site, the risk of hospitalised and uncomplicated malaria per infectious bite over time since discharge and with increasing EIR, and the effect of PDMC on severity of illness. All parameters except for EIR were given uninformative priors (Table S1). We coded the model with a timestep

of one day, with transition rates converted to daily probabilities. The model was fitted using Markov chain Monte Carlo methods in the RStan software[39]. We ran four chains, each having 5000 burn-in and 10,000 sampling iterations. We assessed convergence through visualisation of posterior distributions and the Gelman-Rubin's convergence diagnostic[40]. Full mathematical details of the models, parameters, and fitting procedures are given in the Supplementary Information.

## Population modelling of PDMC demand and impact in different epidemiological settings

We next developed a full population model which tracks SMA in all under five year olds (Figure S3), and included the post-discharge cohort model within this framework (see also Supplementary Information). This deterministic model was similarly coded in discrete time. Children are stratified into two risk groups: those who experienced SMA within the last 6 months (high-risk) and those who did not (low-risk). When an episode of SMA occurs in low-risk individuals, they move to the high-risk state. The fitted post-discharge cohort model placebo arm was used to describe the incidence of malaria episodes in the high-risk group in the absence of PDMC, including SMA episodes. Using estimated total SMA incidence for a given area (see sources in the Results section), the remaining SMA episodes occurring in the low-risk group are calculated. In contrast to the model fitted to the trial setting, we assume the probability of children receiving treatment for uncomplicated episodes or being hospitalised when required is <100% (Table S1). We track SMA cases who are hospitalised separately from those who are not. We vary the assumed probability of hospitalisation from 30–70% based on the recent CARAMAL study tracking children with suspected severe malaria in community settings[24]. We assume that the probability of hospitalisation is not related to previous hospitalisation status. Those who are hospitalised have a fixed 3-day stay in hospital, then we assume that all who survive receive AL at discharge as per standard severe malaria guidelines. Hospitalised SMA cases are eligible for PDMC. PDMC with DP is incorporated as in the post-discharge cohort model, except that adherence is lower as measured in a recent PDMC implementation study[23] (Table S2). We also further stratify the model compartments in children given PDMC to track which courses of PDMC have been taken (any combination of the 1st, 2nd and 3rd courses). After the 6-month high-risk period, individuals return to the low-risk group unless they experience a further episode of SMA. This model thus allows for the iterative process of SMA episodes increasing the risk of future SMA episodes, which affects future PDMC impact and demand. The base case analysis assumes no seasonality in transmission due to uncertainties in the seasonal pattern of SMA cases. To test whether PDMC impact is affected by seasonality, we repeated the analysis assuming that SMA cases follow the same seasonal pattern as EIR (lagged by 15 days to allow for time to symptoms) in each subnational region determined by local rainfall, as established in previous analysis[22,41].

We explicitly model mortality from SMA and from other types of hospitalised malaria during the 25-week post-SMA period. The case fatality rate of SMA in hospital is 7.4% based on a meta-analysis of hospital data[26]. We allow that only a proportion of other malaria inpatients usually meet the WHO severe malaria case definition criteria[27]. Since the severity of patients who are hospitalised may vary between settings, we use an in-hospital case fatality rate of 1% based on country reports from Kenya and Uganda, where the trial was carried out[27,42]. Of note, this is lower than the case fatality rate of ~9% for hospitalised patients who meet the WHO case definition of severe malaria[26]. Case fatality rates outside hospital are highly uncertain. We follow the approach of Camponovo et al.[27] who triangulated verbal autopsy malaria mortality data in the community with severe malaria incidence. In their analysis the case fatality rate of malaria outside hospital is approximately double that within hospital, and therefore we

assume a case fatality rate of 14.8% for non-hospitalised SMA cases and 2% for other cases. In order to keep a constant population size, new individuals are added to the low-risk state at the same rate as the total mortality rate.

The full population model was coded in R and was run to equilibrium, with and without PDMC, to estimate the impact of the intervention. The trials and analyses were approved by the ethics committees at the Kenya Medical Research Institute, Makerere University, the Western Norway Regional Committee for Medical and Health Research Ethics, the Liverpool School of Tropical Medicine, the University of Minnesota, and the Uganda National Council of Science and Technology. Informed consent was originally obtained from parents and guardians.

## Reporting summary

Further information on research design is available in the Nature Portfolio Reporting Summary linked to this article.

## Data availability

The parasite prevalence data are publicly available on the Malaria Atlas Project website https://malariaatlas.org/.[14] Population estimates are available from WorldPop www.worldpop.org.[43] The clinical trial data analysed during the current study will be made available when a research proposal has been approved by the investigators after consideration of overlap between the proposal and any ongoing efforts. Proposals should be directed to feiko.terkuile@lstmed.ac.uk and Bjarne.Robberstad@uib.no; to gain access, data requesters must sign a data access agreement, and the de-identified database will be transferred electronically.

## Code availability

All code is available at https://github.com/lucyokell/pdmc_model[44].

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

## Acknowledgements

The study was funded by the Research Council of Norway through the Global Health and Vaccination (GLOBVAC) Programme (project number 234487), which is part of the European and Developing Countries Clinical Trials Partnership (EDCTP2), supported by the European Union through a cooperative agreement with the Liver-pool School of Tropical Medicine. LCO is funded by a UK Royal Society Dorothy Hodgkin fellowship and acknowledges funding from the MRC Centre for Global Infectious Disease Analysis (reference MR/R015600/1), jointly funded by the UK Medical Research Council (MRC) and the UK Foreign, Commonwealth & Development Office (FCDO), under the MRC/FCDO Concordat agreement and is also part of the EDCTP2 programme supported by the European Union; and acknowledges funding by Community Jameel. MC received support from an award (MR/R010161/1) jointly funded by the UK Medical Research Council (MRC) and the UK Department for International Development (DFID) under the MRC/DFID Concordat agreement, which is also part of the EDCTP2 pro-gramme supported by the European Union and is supported by a Sir Henry Dale Fellowship jointly funded by the Wellcome Trust and the Royal Society (Grant Number 220658/Z/20/Z). A.M. acknowl-edges funding by the UK Medical Research Council (Grant number: G98669, https://mrc.ukri.org/).

## Author contributions

A.T.M. and L.C.O.: Methodology, Software, Formal analysis, Writing—Original Draft, Visualisation, Writing—Review & Editing. M.C.: Con-ceptualisation, Writing—Review & Editing, Funding acquisition. F.O.T.K., K.P., and B.R.: Conceptualisation, Writing—Review & Editing, Project administration, Funding acquisition, Supervision. C.K., T.K.K., A.D., T.N.G., R.O., A.M., R.I., M.J.K., T.C.D.L., and D.J.W.: Data curation, Vali-dation, Investigation, Writing—Review & Editing. P.W., J.D.C.: Methodology.

## Competing interests

The authors declare no competing interests.

## Additional information

**Peer review information** *Nature Communications* thanks Thomas Oba-dia, Olivia Prosper and the other, anonymous, reviewer(s) for their con-tribution to the peer review of this work. Peer reviewer reports are available.

