## [Peer Review File · Nature Communications]

Projected health impact of post-discharge malaria chemoprevention among children with severe malarial anaemia in AfricaREVIEWER COMMENTS

Reviewer #1 (Remarks to the Author):

Review of *Projected health impact of post-discharge malaria chemoprevention among children with severe malarial anaemia in Africa* by
L. Okell *et al.*

Summary: The authors emphasize the benefit of post-discharge malaria chemoprevention (PDMC) in terms of reducing malaria mortality and hospital readmissions in clinical trials in Kenya and Uganda, along with the need to evaluate the efficacy of this control measure in other transmission settings. To this end, they develop a mathematical model of post-discharge malaria, with the local transmission setting as an input, and fit to data from a multi-centre trial of PDMC. The model includes pharmacodynamics of PDMC from earlier work. More specifically, the model was fit to individual patient record data (patients receiving either PDMC or a placebo) using Bayesian methods, and used a semi-informed prior for the entomological inoculation rate (EIR) based on Malaria Atlas Project maps. The main model outputs of interest were the percentage of hospitalized and uncomplicated malaria avoided; the model was able to accurately produce the values derived from the data. The model also tracked the change in daily incidence, allowing the authors to quantify the reduction in hospitalized and uncomplicated malaria cases per infectious mosquito bite. The model was then extended to describe severe malarial anemia (SMA). A key conclusion is that PDMC is most critical in high transmission settings, and because the main benefit of PDMC occurs shortly after discharge, even with imperfect adherence to PDMC, it can remain quite effective. The authors estimate that about 37,000 hospitalizations and over 2000 deaths would be prevented if all hospitalized 0-5 year olds received PDMC, with 3/4 of these accounted for in the 10 countries with the highest malaria burden. Most regions would need fewer than 10 hospitalized children to receive PDMC to prevent one malaria-related death. The authors note an important caveat to these results: namely, that it is difficult to estimate the percentage of individuals who seek hospital care. The authors provide several suggestions to augment the efficacy of a PDMC program, in particular to prolong the protection to 6 months post discharge, including the use of insecticide-treated nets and extending the eligibility to a broader age group.

Comments: This manuscript addresses an important problem with very practical implications and recommendations. The model is supported by different forms of clinical trial data, including a placebo group. The manuscript is, for the most part, clearly written. I provide a few suggestions below to improve the readability that should be straightforward to address. The final comment below is perhaps the most pressing to address, both in a response and in the manuscript itself.

Some minor comments:

1. Fig 1 - PMC should be PDMC for consistency with the text.
2. p. 16 - 2 periods after "episode".

Other comments/questions:

1. Fig 1C - what is the explanation for the discrepancy between the model fits and the data during the first 100 days post discharge?

2. The statement “The incidence of uncomplicated and hospitalised malaria ... per infectious bite decreased” isn’t crystal clear to me. Do the authors mean that for a single individual, the more exposures that individual has, the risk of these outcomes decreases with each additional bite? And if so, is this because of the development of natural immunity?
3. In the beginning of the Methods section, I recommend providing more details about the model structure. The authors describe that the model is a compartmental model, but it is not until the description of the Bayesian inference that the authors state “We coded the model in discrete time with a timestep of one day, with transition rates converted to daily probabilities.” From the supplement, it appears that this model is deterministic, but it is worth noting in the Methods section of the main manuscript. Also, the authors state on p.23 that the full population model is a deterministic discrete-time model; I again suggest placing this statement towards the beginning of the model description.
4. The authors fitted to daily incidence. What likelihood model (or heuristic objective function) was used? I could not find this stated in the main text nor the supplement. The authors used MCMC in the RStan software - is there a default objective function used here? This question is particularly important in the context of Figure 2, which indicates a much smaller uncertainty in the model output than the data suggest, indicating that the likelihood model is not chosen appropriately for the type of data.

Recommendation: I think this manuscript presents important work that I would like to see published in Nature Communications, provided that the comments above (particularly the last one about the uncertainty in model output) are adequately addressed. It is hard to know how extensive this revision will be without knowing the likelihood model used in the model fitting.

Reviewer #2 (Remarks to the Author):

This manuscript contains a mathematical model driven meta-analysis of trial data on post-discharge malaria chemoprophylaxis (PDMC) coupled with a population oriented mathematical model designed to predict the demand and impact of PDMC if delivered effectively throughout sub-Saharan Africa. This work will be important in laying the ground work for building a policy case around the implementation of PDMC in sub-Saharan Africa. It is a novel synthesis of existing trial data and new mathematical modelling and is an important improvement in the translation of trial results for PDMC into planning and policy relevant terms.

The conclusions are in general well supported by the evidence and research undertaken in the work and while additional evidence on cost and operational ability of health systems to deliver PDMC as well as patient adherence to PDMC regimens, the conclusions in terms of population impact and demand seem well grounded in real data and appropriately estimated.

There are no major flaws in the data analysis which I have noted, and the work is clearly written and well edited. One area where the authors might consider additional text/consideration is to highlight some areas of major uncertainty in the predictions of impact where either no data exists or the data that exists is of limited / poor quality. These are especially in the areas of hospital level care seeking among children with SMA, and in the adherence of these children to PDMC regimens in real world settings. Both of these parameters are unknown/incredibly uncertain and both are dramatically important to the estimation of population level effectiveness of these strategies.

Lastly the authors might also consider including information on the relationship of the deaths prevented predicted in their models to the burden of malaria mortality in these areas of SSA. It seems clear but is not explicitly stated that this intervention will never have a major population burden impact, however, it seems to be an incredibly effective way to target resources to a small but extremely vulnerable population and as such might be an incredibly efficient use of resources (where the health system is capable of delivering this with good uptake and adherence).

The authors are to be commended for including their full model code and material necessary to reproduce these analyses for publication.

Reviewer #3 (Remarks to the Author):

This is a very well written manuscript that presents results of models, informed almost exclusively by one multi-center clinical trial, of post-discharge presumptive treatment for malaria in children under 5 years old hospitalized with severe malarial anemia. Note, the authors refer to the intervention as post-discharge malaria chemoprevention, PDMC, although full treatment courses were given at regular intervals. The authors come out strongly in favor of PDMC, but there appear to be some weaknesses in their position.

First, the authors claim that only 2-5 children need to receive PDMC to prevent one malaria re-hospitalization, and fewer than 100 children to prevent one death. This seems implausible, given that the trial that the model uses as its input (Kwambai NEJM 2020) found no significant difference in death between the intervention and placebo arms and actually higher risk of death from any cause in the intervention arm (10% in the PDMC group vs. 4% in the placebo group)-- noted by the trial authors to be consistent with previous studies, which also found higher deaths in the malaria prevention arms. How do the authors of this modeling study reconcile their results with the findings in the source trial, as well as prior trials, which found no significant difference in death, and a likely increase in the risk of death in the chemoprevention arms compared to the placebo arms?

The second major concern is data overfitting. The authors do not explain why the previously validated, "existing well-established" model of malaria transmission failed to predict the clinical trial data well (Lines 358-363). Meanwhile, the authors' customized model contains parameters which have little identifiability (e.g., beta, theta, and epsilon), and the model predictions follow the

actual data almost perfectly, raising concern of overfitting. See figure 1 for example.

Third, the authors do not address the critical issue of drug resistance. The authors do not acknowledge that intermittent presumptive therapy with dihydroartemisinin-piperaquine is essentially monotherapy/monoprophylaxis with piperaquine as the artemisinin component is rapidly eliminated within hours while piperaquine remains in circulation for out to 9 weeks and beyond. Piperaquine resistance was rapid to emerge in parts of SE Asia where the combination was used as first-line (compared to almost all other Plasmodium falciparum-endemic places where artemether-lumefantrine is first line). Could the authors model the emergence of piperaquine resistance in a PDMC vs. non-PDMC situation? The literature seems to have estimates of the parameters needed to populate such a model.

Minor comments:

For Fig 1-panel A, suggest placing the PDMC and placebo side-by-side for each comparison to assist the reader in interpreting the results—i.e., for the UM 3-14 group, show the PDMC next to the placebo, and also suggest adding a third group, the general population of children under 5 years old (shouldn't this important control group be included in all comparisons/charts? They are included, for example, in Fig 2).

Also for Fig 1, did the authors consider the inflection at ~100 days post-discharge in the PDMC group? Once piperaquine is eliminated from the body the PDMC group "catches up" to the non-PDMC group and if the curve were extended beyond the 150 days the authors show the cumulative incidence of hospitalized and uncomplicated cases would likely be no different between the two groups. This is even more evident in the first panel of Fig 1—the PDMC group is no different than the placebo group at 15-25 weeks of follow-up. Meanwhile, there is a very real danger of rendering DP ineffective in short time, with increased drug pressure leading to parasite resistance.

Fig 1: "PMC" should be "PDMC"

Line 275: "in the general population" does this mean the general population in that particular EIR setting, or including settings with different/lower EIR?

Reviewer #1

[Original reviewer comments are in bold.]

Review of Projected health impact of post-discharge malaria chemoprevention among children with severe malarial anaemia in Africa by L. Okell et al.

Summary: The authors emphasize the benefit of post-discharge malaria chemoprevention (PDMC) in terms of reducing malaria mortality and hospital readmissions in clinical trials in Kenya and Uganda, along with the need to evaluate the efficacy of this control measure in other transmission settings. To this end, they develop a mathematical model of post-discharge malaria, with the local transmission setting as an input, and fit to data from a multi-centre trial of PDMC. The model includes pharmacodynamics of PDMC from earlier work. More specifically, the model was fit to individual patient record data (patients receiving either PDMC or a placebo) using Bayesian methods, and used a semi-informed prior for the entomological inoculation rate (EIR) based on Malaria Atlas Project maps. The main model outputs of interest were the percentage of hospitalized

and uncomplicated malaria avoided; the model was able to accurately produce the values derived from the data. The model also tracked the change in daily incidence, allowing the authors to quantify the reduction in hospitalized and uncomplicated malaria cases per infectious mosquito bite.

The model was then extended to describe severe malarial anemia (SMA). A key conclusion is that PDMC is most critical in high transmission settings, and because the main benefit of PDMC occurs shortly after discharge, even with imperfect adherence to PDMC, it can remain quite effective. The authors estimate that about 37,000 hospitalizations and over 2000 deaths would be prevented if all hospitalized 0-5 year olds received PDMC, with 3/4 of these accounted for in the 10 countries with the highest malaria burden. Most regions would need fewer than 10 hospitalized children to receive PDMC to prevent one malaria-related death. The authors note an important caveat to these

results: namely, that it is difficult to estimate the percentage of individuals who seek hospital care. The authors provide several suggestions to augment the efficacy of a PDMC program, in particular to prolong the protection to 6 months post discharge, including the use of insecticide-treated nets and extending the eligibility to a broader age group.

Comments: This manuscript addresses an important problem with very practical implications and recommendations. The model is supported by different forms of clinical trial data, including a placebo group. The manuscript is, for the most part, clearly written. I provide a few suggestions below to improve the readability that should be straightforward to address. The final comment below is perhaps the most pressing to address, both in a response and in the manuscript itself.

Thank you for taking the time to do this thorough and helpful review. We address the comments in detail below.

Some minor comments:

1. Fig 1 - PMC should be PDMC for consistency with the text.

Thank you for spotting this error – we have now altered it to 'PDMC'.

2. p. 16 - 2 periods after \episode".

Now removed.

Other comments/questions:

1. Fig 1C - what is the explanation for the discrepancy between the model fits and the data during the first 100 days post discharge?

Thank you for this question. Figure 1C shows the cumulative daily number of uncomplicated cases in the placebo and PDMC trial arms, comparing data and model fits. The placebo arm model fit is very close to the placebo arm data values. However in the first 100 days, the PDMC trial arm has a slightly higher number of episodes in the data than are predicted by the model fit, as the reviewer comments. The first 100 days are approximately the 14-week (98 days) period during which participants were protected from malaria. This includes 2 weeks of post-treatment prophylaxis following artemether-lumefantrine at discharge, and then an additional 12 weeks by providing 3 courses of PDMC with dihydroartemisinin-piperaquine (DP) at the end of the 2nd, 6th and 10th week post-discharge. Each course of DP provides approximately 4 weeks of post-treatment prophylaxis. The reason for the imperfect fit is that we used fixed parameters to describe the duration and strength of protection provided by the PDMC drug, dihydroartemisinin-piperaquine (DP), from a previous analysis (¹ – see also methods section). We could have chosen to re-estimate the DP drug protection parameters using the PDMC trial data, which would have provided a better fit during these first 100 days. However on balance we felt this could qualify as overfitting of the model, a concern also raised by reviewer 3. We considered the previous analysis which estimated the DP parameters to be robust since it was based on a large dataset containing 8 trial sites, and was able to capture trends well across these varied sites with different transmission. Overall the fit to the current data was close enough that we did not feel justified re-fitting the drug parameters, particularly since our main outcome of interest was prevention of hospitalised episodes, where the model did fit the data well (Figure 1B).

An additional reason for retaining drug parameters from the previous analysis was the diagnostics used for uncomplicated malaria cases during the PDMC trial – some patients were diagnosed with HRP-2 rapid diagnostic tests while others were diagnosed by microscopy. Rapid diagnostic tests are known to have poor predictive value after a recent malaria episode, with individuals remaining positive for around 2 weeks after successful clearance of parasites following treatment of malaria (due to residual antigens rather than live parasites). Similar issues have been encountered in seasonal malaria chemoprevention trials (see reference ² – their Supplementary Table 2). In the PDMC cohort, which was highly exposed to malaria, there may have therefore been some false positives where children had fever from another cause and tested positive by HRP2-based RDT due to a recent rather than current episode of malaria. Although efforts were made during the trial to minimise this effect by confirming infections with microscopy if occurring within 2 weeks of a previous episode, this does not perfectly solve the issue since some individuals remain RDT-positive beyond 2 weeks. This could have slightly overestimated the incidence of uncomplicated malaria. There is insufficient data to estimate this effect in this study, since asymptomatic infection was not tracked during the trial. However we have added an additional explanation of this in the discussion section as follows:

“Some further misclassification could have occurred due to extended circulation of parasite antigens resulting in positive RDTs after successful treatment for a malaria episode. Although efforts were made during the trial to minimise this effect using blood smear confirmation of RDT-positive cases occurring within 2 weeks of a previous episode, another recent study showed the positive predictive value of an RDT relative to microscopy is still under 70% 5 weeks after an episode,³⁶ meaning that the incidence of uncomplicated malaria could be slightly overestimated.”

2. The statement "The incidence of uncomplicated and hospitalised malaria ... per infectious bite decreased" isn't crystal clear to me. Do the authors mean that for a single individual, the more exposures that individual has, the risk of these outcomes decreases with each additional bite? And if so, is this because of the development of natural immunity?

We model this effect as a population average reduction in the probability of symptomatic malaria as EIR increases, rather than an individual increase in immunity with accumulated exposure, and we have now clarified the wording in the main text to reflect this. As the EIR increases, the incidence of symptomatic malaria (uncomplicated and hospitalised) has been observed to increase non-linearly, so that the incidence 'per bite' is lower.⁴ It may indeed result from the more rapid development of infection-blocking and anti-disease immunity in areas of high transmission. It may also result from density-dependent effects whereby incoming infections compete with one another and cannot all establish in the host simultaneously. Infectious bites close together in time may also simply create one symptomatic episode rather than several. We use one empirical function to capture these possible effects, taking a similar functional form to that used in previous analyses.^{5,6}

In the methods we edited the text as follows: "The *average* probability that an infectious bite leads to infection and symptoms are both known to decline with EIR in the general population *due to acquired immunity and density-dependent effects, e.g. competition between parasites*. We included this possibility by further scaling *total incidence in each location* by a similar functional form to that identified in previous analyses⁷"

In the results we added: "The incidence of uncomplicated and hospitalised malaria increased with EIR, but the *average* risk of these outcomes per infectious bite decreased"

3. In the beginning of the Methods section, I recommend providing more details about the model structure. The authors describe that the model is a compartmental model, but it is not until the description of the Bayesian inference that the authors state \We coded the model in discrete time with a timestep of one day, with transition rates converted to daily probabilities." From the supplement, it appears that this model is deterministic, but it is worth noting in the Methods section of the main manuscript. Also, the authors state on p.23 that the full population model is a deterministic discrete-time model; I again suggest placing this statement towards the beginning of the model description.

We have now moved this model description to the beginning of the methods: "We developed a deterministic, discrete-time compartmental model.."

And a similar sentence at the beginning of the description of the full population model: "*This deterministic model was similarly coded in discrete time.*"

4. The authors fitted to daily incidence. What likelihood model (or heuristic objective function) was used? I could not find this stated in the main text nor the supplement. The authors used MCMC in the RStan software - is there a default objective function used here? This question is particularly important in the context of Figure 2, which indicates a much smaller uncertainty in the model output than the data suggest, indicating that the likelihood model is not chosen appropriately for the type of data.

Thank you for noticing this important omission – you are correct that we had not stated the likelihood model in the supplement and it was only available in the online code. We have now added a full description in the supplement (also copied below). In our original submission, we had modelled the daily incidence of uncomplicated or hospitalised malaria on a given day as Poisson distributed. The reviewer's comment prompted us to re-examine the likelihood model in more detail. We have carried out further analysis and found that negative binomial likelihood gave a better fit to the data, as assessed by visual inspection of the posterior predictive checks (PPC) and the log-likelihood. We have now switched to using negative binomial likelihood instead and we have added a supplementary figure S8 showing the PPC of incidence in each site compared with the data values in each site as shown in Figure 2 (also copied below here). The figures show that the PPC and its

uncertainty adequately encompass the majority of observed data. We thank the reviewer for identifying this issue which has improved the analysis.

The effect of changing the likelihood model on the posterior central estimates and distributions of key model parameters was not substantial (see comparison Table below) and therefore once propagated through to the rest of the results, did not produce any large change in the quantitative results in the manuscript (other than the posterior predictive intervals in Figure S8). The uncertainty in Figure 2 which the reviewer raises, remained similar. To clarify, the credible interval shown in this figure is the uncertainty around the mean prediction, rather than showing a posterior predictive interval, which we have now made clearer by revising the Figure 2 legend and referring the reader to the posterior predictive checks in the supplement. We also now plot the posterior EIR on the x-axis of Figure 2 and added horizontal 95% CrI to better show the uncertainty in EIR (also copied here).

Changes in the manuscript:

We have now added the following text to describe the likelihood model in the Supplement.

Model fitting was undertaken using MCMC in the RStan software.⁸ The expected number W of children experiencing an event of type q was modelled each day, with the 4 possible event types being uncomplicated cases in the placebo group, uncomplicated cases in the PDMC group, hospitalized cases in the placebo group, and hospitalized cases in the PDMC group.

The expected number of uncomplicated malaria cases on day t_d in a given site in the placebo group (event type $q=1$) was calculated as:

$$W_1(t_d) = y_d(t_d - 1)e_{AL}S(t_d - 1)(1 - \theta_1)$$

and in the PDMC group (event type $q=2$) as:

$$W_2(t_d) = y_d(t_d - 1)e_{AL}S'(t_d - 1)(1 - \theta)(1 - p_{pDP}(t_{DP})p_{ad})$$

while the daily expected number of hospitalised malaria cases was calculated in the placebo group (event type $q=3$) as:

$$W_3(t_d) = y_d(t_d - 1)e_{AL}S(t_d - 1)(1 - \theta_1)$$

and in the PDMC group (event type $q=4$) as:

$$W_4(t_d) = y_d(t_d - 1)e_{AL}S'(t_d - 1)(1 - \theta)(1 - p_{pDP}(t_{DP})p_{ad})$$

The daily observed number of events Y of type q on day t_d , using the same annotation, are assumed to follow a negative binomial distribution with mean W_q and shape k :

$$Y_q(t_d) \sim \text{Negative binomial}(W_q(t_d), k)$$

Additional supplementary figures: posterior predictive checks.

Figure S8. Posterior predictive checks of the model against data shown in Figure 2. Incidence of uncomplicated (left) and hospitalized (right) malaria in the placebo arm in weeks 3-14 post-discharge by trial hospital: data (red) with 95% CI, and the model (blue) with posterior predictive interval

calculated using 1000 samples from the joint posterior distribution including the model-estimated variance.

Revised Figure 2 and legend:

Figure 2 (A) Relationship between posterior estimated EIR values and incidence of uncomplicated malaria per 100 person-years in 0–5-year-olds 3-14 weeks after hospital discharge following severe malarial anaemia. The data are from the 9 trial locations in Uganda and Kenya (red; data post-discharge (trial) = data from Kwambai et al.11 Vertical error bars = 95% confidence interval of the incidence data; horizontal error bars = 95% credible interval of posterior EIR estimate). The fitted model is in blue (shaded area = 95% credible interval around the mean). For comparison, the estimated incidence of uncomplicated malaria by EIR in the general population of under five-year-olds is shown in black. (B) As (A) showing hospitalised malaria episodes in the same trial group (log scale). For comparison, the estimated incidence of hospitalised malaria in the general population of children aged 0-5 years for a given annual EIR is shown in black. Three additional validation data points are shown from separate post-discharge studies (purple; Uganda: Opoka et al.17,18 and Malawi: Phiri et al.10) which were not used for fitting the model. Posterior predictive checks of the model against the data in each site are shown in Figure S8.

Comparison table: posterior parameter estimates and 95% credible intervals under the original Poisson likelihood model, compared to the revised model using negative binomial likelihood.

Parameter	Description	Poisson model Posterior value (95% CrI)	Negative binomial model Posterior value (95% CrI)
EIR	Annual EIR in adults for each trial site		
	Siaya	28.4 (23.7, 34.2)	28.3 (23.5, 33.9)
	Kisumu	5.4 (4.5, 6.4)	5.4 (4.4, 6.4)
	Homa Bay	10.9 (8.7, 13.3)	11.1 (8.9, 13.7)
	Migori	2.5 (2.1, 2.9)	2.5 (2.1, 2.9)
	Jinja	18.5 (16.0, 21.2)	18.1 (15.5, 20.9)
	Hoima	5.5 (4.5, 6.6)	5.6 (4.6, 6.7)
	Masaka	5.2 (4.3, 6.1)	5.1 (4.3, 6.1)
	Mubende	5.3 (4.5, 6.4)	5.3 (4.4, 6.3)
	Kamuli	24.5 (18.1, 34.5)	25.7 (18.6, 35.4)
$b\phi\xi$	Maximum relative incidence of symptomatic malaria in post discharge cohort relative to EIR in adults	1.00 (0.61, 1.84)	1.02 (0.62, 1.86)
w	Rate parameter for the relationship of EIR with probability of symptomatic malaria post-discharge	0.026 (0.011, 0.040)	0.028 (0.013, 0.042)
θ_1	Probability of requiring hospitalisation post-discharge among	0.39 (0.36, 0.42)	0.38 (0.35, 0.42)

	symptomatic malaria cases, not under active PDMC protection*		
θ_2	Probability of requiring hospitalisation post-discharge among symptomatic malaria cases, under active PDMC protection*	0.25 (0.17, 0.34)	0.24 (0.16, 0.33)
λ_{risk}	Scale parameter describing the decline in incidence over time post discharge	108.4 (27.3, 186.1)	109.7 (28.8, 189.6)
η_{risk}	Shape parameter describing the decline in incidence over time post discharge	0.64 (0.34, 1.29)	0.65 (0.35, 1.32)
k	Shape parameter of the negative binomial distribution of daily incidence	-	3.90 (2.16, 8.59)

Recommendation: I think this manuscript presents important work that I would like to see published in Nature Communications, provided that the comments above (particularly the last one about the uncertainty in model output) are adequately addressed. It is hard to know how extensive

this revision will be without knowing the likelihood model used in the model fitting.

We hope we have now clarified the likelihood model.

Reviewer #2 (Remarks to the Author):

This manuscript contains a mathematical model driven meta-analysis of trial data on post-discharge malaria chemoprophylaxis (PDMC) coupled with a population oriented mathematical model designed to predict the demand and impact of PDMC if delivered effectively throughout sub-Saharan Africa. This work will be important in laying the ground work for building a policy case around the implementation of PDMC in sub-Saharan Africa. It is a novel synthesis of existing trial data and new mathematical modelling and is an important improvement in the translation of trial results for PDMC into planning and policy relevant terms.

The conclusions are in general well supported by the evidence and research undertaken in the work and while additional evidence on cost and operational ability of health systems to deliver PDMC as well as patient adherence to PDMC regimens, the conclusions in terms of population impact and demand seem well grounded in real data and appropriately estimated.

We thank the reviewer for these positive comments.

There are no major flaws in the data analysis which i have noted, and the work is clearly written and well edited. One area where the authors might consider additional text/consideration is to highlight some areas of major uncertainty in the predictions of impact where either no data exists or the data that exists is of limited / poor quality. These are especially in the areas of hospital level care seeking among children with SMA, and in the adherence of these children to PDMC regimens in real world settings. Both of these parameters are unknown/incredibly uncertain and both are dramatically important to the estimation of population level effectiveness of these strategies.

We agree these are both extremely important uncertainties and we have added further text to the discussion to emphasise and address these as follows (underlined):

“Our results on the number of cases averted per child given PDMC and the number needed to treat to avert uncomplicated and hospitalised episodes were derived from clinical trial results and are relatively robust to model assumptions. However, the total demand for PDMC and the total number of cases and deaths that could be prevented are uncertain, given that PDMC is a hospital-based intervention and there is a lack of data on the probability of accessing hospital care. Access to hospitals is likely to vary greatly between and within countries.²⁸ A recent study tracking children with suspected severe malaria in the community in Uganda, Nigeria, and DRC found that only 41-65% access hospital care even after referral by a community health worker.²⁵ **There are similar findings in LMIC settings relating to other diseases; for example in Lusaka, Zambia, only 32% of fatal respiratory syncytial virus cases in infants sought care in hospital before death. In our analysis we varied the percentage who access hospital care from 30-70% to allow for this uncertainty, however it is possible that in some countries, access is lower or higher, meaning that we over- or underestimate both demand and impact of PDMC.**”

We added a further paragraph to the discussion to address the reviewer’s comment about potential adherence to PDMC during implementation:

“Our results are also sensitive to the assumed level of adherence to PDMC during routine implementation, which we based on an implementation trial that simply provided all treatments at discharge without further reminders.²⁴ 77-89% adherence was observed (adherence varied across the 1st-3rd treatments). If lower adherence was achieved during routine implementation, we could over-estimate impact of PDMC. However, high adherence has generally been observed in other malaria chemoprevention interventions. For example routine seasonal malaria chemoprevention programmes in the Sahel region report adherence of 87-99%.³² “

Lastly the authors might also consider including information on the relationship of the deaths prevented predicted in their models to the burden of malaria mortality in these areas of SSA. It seems clear but is not explicitly stated that this intervention will never have a major population burden impact, however, it seems to be incredibly effective way to target resources to a small but extremely vulnerable population and as such might be an incredibly efficient use of resources (where the health system is capable of delivering this with good uptake and adherence).

We agree with the reviewer’s assessment on the population impact and mortality burden. We have added the following to the discussion to acknowledge this.

“Given that the vulnerable group targeted by PDMC is small, we estimate PDMC would prevent only a fraction of total population SMA cases (5-9% in high transmission areas) and a smaller fraction of all malaria deaths (WHO estimates ~427,200 malaria deaths in under fives in 2019). This highlights the importance of implementing PDMC alongside key preventive interventions across the whole community”

The authors are to be commended for including their full model code and material necessary to reproduce these analyses for publication.

Reviewer #3 (Remarks to the Author):

This is a very well written manuscript that presents results of models, informed almost exclusively by one multi-center clinical trial, of post-discharge presumptive treatment for malaria in children under 5 years old hospitalized with severe malarial anemia. Note, the authors refer to the intervention as post-discharge malaria chemoprevention, PDMC, although full treatment courses were given at regular intervals. The authors come out strongly in favor of PDMC, but there appear to be some weaknesses in their position.

First, the authors claim that only 2-5 children need to receive PDMC to prevent one malaria re-hospitalization, and fewer than 100 children to prevent one death. This seems implausible, given that the trial that the model uses as its input (Kwambai NEJM 2020) found no significant difference in death between the intervention and placebo arms and actually higher risk of death from any cause in the intervention arm (10% in the PDMC group vs. 4% in the placebo group)-- noted by the trial authors to be consistent with previous studies, which also found higher deaths in the malaria prevention arms. How do the authors of this modeling study reconcile their results with the findings in the source trial, as well as prior trials, which found no significant difference in death, and a likely increase in the risk of death in the chemoprevention arms compared to the placebo arms?

Thank you for raising these concerns, we agree it is important to clarify these points. The author suggests the mortality was higher in the PDMC than control arm in the previous trial by Kwambai et al. This is a misunderstanding. During the intervention period, there was a 94% (95% CI 47-99) reduction in mortality by PDMC compared to placebo, which was statistically significant. During the extended follow-up period post-intervention, when protective drug levels had waned, the effect was in the opposite direction (not statistically significant). Still, overall the cumulative effect by 6 months (i.e. combining the intervention period and the extended follow-up period after protective drug levels had waned) was in favour of PDMC (protective efficacy = 35%, -38, to 70), but not significant. The trial was not intended/powerd to detect mortality impact.

To be clear, the result that “2-5 children need to receive PDMC to prevent one malaria re-hospitalization, and fewer than 100 children to prevent one death” relates to high burden endemic areas, not all endemic areas, as indicated in the abstract (the 20 highest burden countries). In our analysis we did not use mortality results from the trial given the low numbers. In addition, we model only malaria-related mortality while the trial results showed all-cause mortality. We assumed that case fatality rates amongst those post-discharge would be similar as in other severe malaria cases, which have been established by larger studies of malaria mortality. Nonetheless our predictions are broadly in the same range as impact in the trial, given that 525 children were treated with PDMC, 4 deaths were averted, and therefore 131 children needed to receive PDMC to avert 1 episode.

We agree that it is good to acknowledge the uncertainty over mortality impact and clarify the results of the original trial. We have added the following additional points to our discussion:

“Mortality impact of PDMC remains a topic to be studied further in future, given that PDMC trials to date have not been powered to evaluate impact on mortality. Overall, 3 months PDMC with DP was

associated with a large drop in all-cause mortality in the trial of 94% during the 3-month intervention period, but over the whole 6 months follow up there was a non-significant 35% reduction in mortality (95% CI -38, 70%).”

The result that 2-5 children need to receive PDMC to prevent one hospitalised malaria episode is also consistent with the original trial. The trial data overall across all sites with varying transmission shows a number needed to treat (NNT) of 4 with respect to hospitalized malaria (524 children got PDMC. There were 316 readmissions/deaths in the placebo group and 184 in the PDMC group, so an estimated 132 episodes were prevented among the 524. $NNT = 524/132 = 4$). In one of the higher transmission sites with most cases, Jinja, the NNT was 1.4 (97 hospitalized cases averted and 133 children treated). Our modelling results match the trial data and further characterise how the impact varies with transmission intensity to project impact in other sites.

The second major concern is data overfitting. The authors do not explain why the previously validated, “existing well-established” model of malaria transmission failed to predict the clinical trial data well (Lines 358-363). Meanwhile, the authors’ customized model contains parameters which have little identifiability (e.g., beta, theta, and epsilon), and the model predictions follow the actual data almost perfectly, raising concern of overfitting. See figure 1 for example.

The reviewer raises a good point that we did not explain why the existing well-established model could not predict the clinical data. We have now added some text into the methods to provide a better explanation. The reason is that there is no mechanism in that previously developed model allowing for additional vulnerability following one episode of SMA, indeed the model assumes that experiencing one episode will increase immunity and reduce the probability of a future episode. The established model was fitted to population-level incidence of hospitalised malaria and not any longitudinal data on risk within individuals over time, which is why this phenomenon was not known or included. Indeed the trial and other post-discharge studies have provided valuable data suggesting that the model should be changed to account for this high risk period after SMA. We added a sentence to the text as follows (new text underlined).

“preliminary analysis indicated that the high incidence of uncomplicated and hospitalised malaria after a severe malarial anaemia (SMA) episode, as observed in trials and clinical studies,¹⁵ was not adequately captured using this existing model, despite allowing for individual variation in immunity and exposure to mosquito bites. This is because the model was calibrated against population level incidence data, not against longitudinal data on risk within individuals over time. For the current analysis, a novel model was created to describe the natural history of malaria illness in the post-discharge population, how it changes over time and with varying malaria transmission intensities.”

We aimed to avoid overfitting the data by keeping the model as simple as possible and informed by previous work as well as the trial data. For example, we did not re-fit the drug duration of protection parameters, instead fixing them to previously established values. While Figure 1 shows a relatively good fit, it is not perfect, especially in the first 100 days in the PDMC arm as queried by reviewer #1 (see above). Figure 2 and our posterior predictive checks in Figure S8 show a large uncertainty in predictions by EIR which is appropriately captured by the likelihood model we selected, although the point estimates show an imperfect fit.

Most previous models have had separate parameters for zeta, b, phi ($\xi b \phi$), which model the process leading to a case of symptomatic malaria (we think the reviewer is referring to these 3 parameters in this comment). They describe the relative probability of exposure to an infectious bite, the probability of an infectious bite leading to a blood-stage malaria infection, and the probability that the infection then becomes symptomatic, respectively. Only symptomatic malaria was directly assessed during this

trial, not infection or exposure to bites. Recognising this, we combined these 3 parameters effectively into 1 parameter, (incidence of symptomatic malaria) which is identifiable in the trial. We retained the standard notation for comparability with previous work and for clarity to other malaria modellers. We also used the parameter theta in the model to indicate the proportion of all symptomatic cases which require hospitalisation, which is identifiable in the trial since uncomplicated and hospitalised cases were recorded.

Third, the authors do not address the critical issue of drug resistance. The authors do not acknowledge that intermittent presumptive therapy with dihydroartemisinin-piperazine is essentially monotherapy/monoprophylaxis with piperazine as the artemisinin component is rapidly eliminated within hours while piperazine remains in circulation for out to 9 weeks and beyond. Piperazine resistance was rapid to emerge in parts of SE Asia where the combination was used as first-line (compared to almost all other Plasmodium falciparum-endemic places where artemether-lumefantrine is first line). Could the authors model the emergence of piperazine resistance in a PDMC vs. non-PDMC situation? The literature seems to have estimates of the parameters needed to populate such a model.

The reviewer is right, we had not mentioned drug resistance and it is good to raise this important issue. Piperazine resistance did indeed emerge in areas where the treatment was used as first line therapy. The key difference in using DP for PDMC versus first-line therapy is the proportion of parasites likely to be exposed to the drug. PDMC targets a very select group – children under 5 with severe malarial anaemia, a relatively rare outcome of malaria infection. In areas with widespread piperazine resistance, namely Cambodia, the policy was to treat all uncomplicated cases in all age groups with DP. The selection pressure on the parasite for evolution and spread of resistance created by PDMC is likely to be minimal compared to its use as first line therapy. As an example, Cameroon reported ~3 million malaria cases seeking care in 2020, whereas we estimate that only 4,300 children (~0.2% of these) would require PDMC per year. For a study on the relationship between drug consumption and selection of resistance see the reference by Watson et al ¹⁶.

Some countries have begun to use DP as one of several first-line treatments – as of the WHO malaria report in December 2021 this included Cameroon, Ghana and Nigeria. The selection for resistance created by using DP for uncomplicated malaria cases in these settings in Africa is likely to dwarf the use of DP for PDMC.

Another key difference between the use of DP for first line therapy versus PDMC of relevance to drug resistance is that children have already had their parasites cleared by AL treatment in hospital before receiving chemoprevention. This reduces the selection pressure exerted by DP since the parasites from the initial infection are not exposed to the drug. Only new infections during follow up are exposed to DP.

We agree it would be very interesting to model the emergence of piperazine resistance and explore these issues using a mathematical model. The type of model required to explore this question is substantially different from the methods used here, since we would need a dynamic framework including multiple malaria strains, community treatment of uncomplicated malaria and relevant parameters of resistant strains (fitness costs, survival advantage etc). Some of the authors on this paper are currently working on such model developments (e.g. ¹⁶) and we hope in future to investigate this important question of whether chemoprevention could trigger resistance development. This is a large undertaking beyond the scope of this paper. We have however added additional discussion of drug resistance in the paper to address the reviewer's comment as follows:

“Resistance to piperazine has emerged in the Greater Mekong region, where DP was used for first line therapy of uncomplicated malaria. There are concerns that use of DP for chemoprevention could lead to the emergence of piperazine resistance in Africa. However, the very small proportion of parasites that would be exposed to DP through PDMC programmes likely lowers the risk of resistance considerably. For example, in Nigeria, the estimated number of children eligible for PDMC each year, 41,400, is only ~0.1% of total malaria cases reported by the country to WHO.⁵ Introduction of DP as a first-line treatment option in a number of African countries (Cameroon, Ghana and Nigeria⁵) is likely to substantially outweigh the use of DP for PDMC in terms of selecting for resistance. “

Minor comments:

For Fig 1-panel A, suggest placing the PDMC and placebo side-by-side for each comparison to assist the reader in interpreting the results—i.e., for the UM 3-14 group, show the PDMC next to the placebo, and also suggest adding a third group, the general population of children under 5 years old (shouldn't this important control group be included in all comparisons/charts? They are included, for example, in Fig 2).

We like the reviewer's suggestion to place the PDMC and placebo group next to one another in Figure 1A, and have revised the figure to do so (copied below). We agree this helps make the comparison easier for the reader. We have not included the general population in this figure since we would like the reader to focus on the comparison of model and trial data in this figure, and we do not have data on incidence in the general population from the trial areas. The main reason for including the general population in Fig 2 is to compare their incidence with the very high incidence in children post-discharge after SMA. This result highlights the vulnerability of the post-discharge group to malaria. In later figures and charts we include the full population of under fives with respect to their SMA incidence, PDMC demand and PDMC impact (Figure 3, Table 1).

Figure 1: Model fits to trial data.

Comparison of the number of malaria cases from the trial and model predictions across all sites. (A) Hospitalised (left) and uncomplicated malaria (right) in weeks 3-14 and 15-25 post discharge in PDMC (blue) and placebo (black) trial arms. Error bars show 95% credible intervals (CI) of model predictions (B) Cumulative daily number of hospitalised malaria cases by time since discharge after the original SMA episode in placebo (black) and PDMC (blue) trial arms; solid line=data, dashed line and shaded area=model fit and 95% CI. (C) As B, for uncomplicated malaria cases.

Also for Fig 1, did the authors consider the inflection at ~100 days post-discharge in the PDMC group? Once piperazine is eliminated from the body the PDMC group “catches up” to the non-PDMC group and if the curve were extended beyond the 150 days the authors show the cumulative incidence of hospitalized and uncomplicated cases would likely be no different between the two groups. This is even more evident in the first panel of Fig 1—the PDMC group is no different than the placebo group at 15-25 weeks of follow-up. Meanwhile, there is a very real danger of rendering DP ineffective in short time, with increased drug pressure leading to parasite resistance.

We agree that the incidence in the PDMC group is no different from the placebo group in weeks 15-25 of follow up as shown in Figure 1. This is exactly what we expect since the PDMC drugs are no longer active in this time period, starting at week 15, since the last course of PDMC with DP is given at 10 weeks post-discharge. The equality between trial arms during this non-intervention period demonstrates the effectiveness of the randomisation procedure.

We disagree, however, that the PDMC group number of cases would ‘catch up’ with the placebo group if the follow up were extended. On the contrary, the equal number of cases between arms after the intervention ended shows that there is no catch up – if there were then the cases in the PDMC arm would need to be higher than the placebo arm. The risk in each group during this time is the same and not affected by the prior intervention.

Fig 1: “PMC” should be “PDMC”

Thank you, this has now been changed.

Line 275: “in the general population” does this mean the general population in that particular EIR setting, or including settings with different/lower EIR?

Yes the former is correct. We have now added to the text “EIR in the general population in any given setting” to clarify.

Other changes

*Since we originally submitted this manuscript, the World Health Organisation has adopted PDMC as a policy recommendation as of 3rd June 2022.¹⁷ The new policy recommends “Children admitted to hospital with severe anaemia living in settings with moderate to high malaria transmission should be given a full therapeutic course of an antimalarial medicine at predetermined times following discharge from hospital to reduce re-admission and death.... Moderate to high perennial malaria transmission settings are defined as areas with a *P. falciparum* parasite prevalence greater than 10%”*

We therefore refined our manuscript slightly to reflect this, in particular focussing results section to show impact of PDMC in areas which have a parasite prevalence of >10%. The following changes were made:

In the introduction section we added a sentence: “In 2022, the World Health Organisation adopted PDMC in the malaria guidelines, recommending its use in areas with parasite prevalence >10%.¹²”

In the results section, we revised Figure 3 to show the number of hospitalized episodes averted per child given PDMC in relation to parasite prevalence (Fig 3C). We additionally reported some results:

Model update: since writing our manuscript we were asked by colleagues to test whether our model results varied if seasonal variation was included in the model. We have now added a short sensitivity analysis to the supplement and a brief mention in the main text (overall results were not affected). We also updated population estimates using UN population projections for 2019.

References

- 1 Okell LC, Cairns M, Griffin JT, *et al.* Contrasting benefits of different artemisinin combination therapies as first-line malaria treatments using model-based cost-effectiveness analysis. *Nat Commun* 2014; **5**: 5606.
- 2 Phiri MD, Cairns M, Zongo I, *et al.* The duration of protection from azithromycin against malaria, acute respiratory, gastrointestinal, and skin infections when given alongside seasonal malaria chemoprevention: Secondary analyses of data from a clinical trial in houndé, Burkina Faso, and bougouni, Mali. *Clin Infect Dis* 2021; **73**: e2379–86.
- 3 Cameron E, Battle KE, Bhatt S, *et al.* Defining the relationship between infection prevalence and clinical incidence of Plasmodium falciparum malaria. *Nat Commun* 2015; **6**: 8170.
- 4 Griffin JT, Ferguson NM, Ghani AC. Estimates of the changing age-burden of Plasmodium falciparum malaria disease in sub-Saharan Africa. *Nat Commun* 2014; **5**: 3136.
- 5 Griffin JT, Bhatt S, Sinka ME, *et al.* Potential for reduction of burden and local elimination of malaria by reducing Plasmodium falciparum malaria transmission: a mathematical modelling study. *Lancet Infect Dis* 2016; **16**: 465–72.
- 6 Griffin JT, Hollingsworth TD, Okell LC, *et al.* Reducing Plasmodium falciparum malaria transmission in Africa: a model-based evaluation of intervention strategies. *PLoS Med* 2010; **7**. DOI:10.1371/journal.pmed.1000324.
- 7 Stan Development Team. RStan: the R interface to Stan. R package version 2.21.2, <http://mc-stan.org/>. 2020.
- 8 Kwambai TK, Dhabangi A, Idro R, *et al.* Malaria chemoprevention in the postdischarge management of severe anemia. *N Engl J Med* 2020; **383**: 2242–54.
- 9 Opoka RO, Hamre KES, Brand N, Bangirana P, Idro R, John CC. High postdischarge morbidity in Ugandan children with severe malarial anemia or cerebral malaria. *J Pediatric Infect Dis Soc* 2016; : iw060.
- 10 Opoka RO, Waiswa A, Harriet N, John CC, Tumwine JK, Karamagi C. Blackwater fever in Ugandan children with severe anemia is associated with poor postdischarge outcomes: A prospective cohort study. *Clin Infect Dis* 2020; **70**: 2247–54.
- 11 Phiri K, Esan M, van Hensbroek MB, Khairallah C, Faragher B, ter Kuile FO. Intermittent preventive therapy for malaria with monthly artemether-lumefantrine for the post-discharge management of severe anaemia in children aged 4-59 months in southern Malawi: a multicentre, randomised, placebo-controlled trial. *Lancet Infect Dis* 2012; **12**: 191–200.
- 12 Camponovo F, Bever CA, Galactionova K, Smith T, Penny MA. Incidence and admission rates for severe malaria and their impact on mortality in Africa. *Malar J* 2017; **16**: 1.
- 13 Hetzel MW, Okitawutshu J, Tshetu A, *et al.* Effectiveness of rectal artesunate as pre-referral treatment for severe malaria in children <5 years of age. bioRxiv. 2021; : 2021.09.24.21263966.
- 14 Kwambai TK, Dhabangi A, Idro R, *et al.* Malaria chemoprevention with monthly dihydroartemisinin-piperazine for the post-discharge management of severe anaemia in

children aged less than 5 years in Uganda and Kenya: study protocol for a multi-centre, two-arm, randomised, placebo-controlled, superiority trial. *Trials* 2018; **19**: 610.

- 15 Boni MF, Smith DL, Laxminarayan R. Benefits of using multiple first-line therapies against malaria. *Proc Natl Acad Sci U S A* 2008; **105**: 14216–21.
- 16 Watson OJ, Gao B, Nguyen TD, *et al.* Pre-existing partner-drug resistance facilitates the emergence and spread of artemisinin resistance: a consensus modelling study. bioRxiv. 2021; : 2021.04.08.437876.
- 17 World Health Organization. WHO Guidelines for malaria. 2022; published online June 3. <https://www.who.int/publications/i/item/guidelines-for-malaria> (accessed June 28, 2022).

Review of the manuscript “Projected health impact of post-discharge malaria chemoprevention among children with severe malarial anaemia in Africa” by Okell *et al.*

Reviewer: Dr. Thomas Obadia

In their manuscript, Okell *et al.* estimate the impact of post-discharge malaria chemoprevention (PDMC) after severe malarial anaemia (SAM). The key findings highlight the substantial benefits of PDMC with low numbers of children treated required to reduce the number of hospital readmissions and subsequent deaths. While not dwelling into actual cost-effectiveness analyses, it is undeniable that these results will be useful to policymakers to design and optimize public health control measures.

The manuscript models the outcomes of interest in a two-step process, with both steps relying on deterministic, compartmental models in a Bayesian setting. As a first step, the authors model the PDMC process in a longitudinal framework, with input being transmission (entomological infectious rate, EIR) at the locations from a multicenter clinical trial (PDMC vs. placebo). The outputs of interest are uncomplicated new malaria episodes and hospital readmissions. The model was able to accurately reproduce the trend between EIR and outcomes of interest. All 9 centers from the trial were used in that PDMC model while 3 additional trials (in Uganda and Malawi, conducted by co-authors of the present manuscript) were used as validation to assess model ability to predict incidence. In a second step, the authors embedded their PDMC model into a larger model of SMA in sub-Saharan Africa. This second model is not extensively described in the main text, but rather in the Supplementary Material: it essentially uses the same Bayesian framework and children, initially from the general population (G), can develop SMA and either go through hospital admission (H) and associated pathways (artemeter-lumefantrine prophylaxis, followed or not by PDMC as modeled before) or have their disease handled in the community only (Dc). Any SMA episode, regardless of its pathway (hospital or community) is followed by a susceptible period at high risk and another one at lower risk (subscript nHR). Recurrent SMA is defined as any transition from high risks compartment to a new SMA episode. Children may also not experience SMA in the following 6 months and they then return to the G state.

The key findings are well supported by the model and data, the scientific approach seems sound and the author clearly discuss the limitations of their modelling approach in the discussion, in particular with respect to generalization of some results due to uncertainty in key parameters like treatment-seeking behavior and adherence. The source code for the model has also been uploaded to a GitHub repository: this is an excellent initiative and while

the repository would benefit from a README and short description to ease reproducibility, it is still rare enough of an initiative to be highlighted.

My review will build upon the comments from Reviewer #1 and Reviewer #2 and subsequent answers from the authors, as requested by the handling Editor. In the following, the original reviewer comments and answers will be in separate boxes, and my comments will follow.

I am overall impressed with the amount of work and the writing of this manuscript, and I would like to see it published in Nature Communications as it will benefit its readership. I do however still have concerns that need to be addressed before publication can be recommended.

Reviewer #1

The only 2 minor comments have been properly addressed.

1. Fig 1C - what is the explanation for the discrepancy between the model fits and the data during the first 100 days post discharge?

Thank you for this question. Figure 1C shows the cumulative daily number of uncomplicated cases in the placebo and PDMC trial arms, comparing data and model fits. The placebo arm model fit is very close to the placebo arm data values. However in the first 100 days, the PDMC trial arm has a slightly higher number of episodes in the data than are predicted by the model fit, as the reviewer comments. The first 100 days are approximately the 14-week (98 days) period during which participants were protected from malaria. This includes 2 weeks of post-treatment prophylaxis following artemether-lumefantrine at discharge, and then an additional 12 weeks by providing 3 courses of PDMC with dihydroartemisinin-piperaquine (DP) at the end of the 2nd, 6th and 10th week post-discharge. Each course of DP provides approximately 4 weeks of post-treatment prophylaxis. The reason for the imperfect fit is that we used fixed parameters to describe the duration and strength of protection provided by the PDMC drug, dihydroartemisinin-piperaquine (DP), from a previous analysis (¹ – see also methods section). We could have chosen to re-estimate the DP drug protection parameters using the PDMC trial data, which would have provided a better fit during these first 100 days. However on balance we felt this could qualify as overfitting of the model, a concern also raised by reviewer 3. We considered the previous analysis which estimated the DP parameters to be robust since it was based on a large dataset containing 8 trial sites, and was able to capture trends well across these varied sites with different transmission. Overall the fit to the current data was close enough that we did not feel justified re-fitting the drug parameters, particularly since our main outcome of interest was prevention of hospitalised episodes, where the model did fit the data well (Figure 1B).

An additional reason for retaining drug parameters from the previous analysis was the diagnostics used for uncomplicated malaria cases during the PDMC trial – some patients were diagnosed with HRP-2 rapid diagnostic tests while others were diagnosed by microscopy. Rapid diagnostic tests are known to have poor predictive value after a recent malaria episode, with individuals remaining positive for around 2 weeks after successful clearance of parasites following treatment of malaria (due to residual antigens rather than

live parasites). Similar issues have been encountered in seasonal malaria chemoprevention trials (see reference ² – their Supplementary Table 2). In the PDMC cohort, which was highly exposed to malaria, there may have therefore been some false positives where children had fever from another cause and tested positive by HRP2-based RDT due to a recent rather than current episode of malaria. Although efforts were made during the trial to minimise this effect by confirming infections with microscopy if occurring within 2 weeks of a previous episode, this does not perfectly solve the issue since some individuals remain RDT-positive beyond 2 weeks. This could have slightly overestimated the incidence of uncomplicated malaria. There is insufficient data to estimate this effect in this study, since asymptomatic infection was not tracked during the trial. However we have added an additional explanation of this in the discussion section as follows:

“Some further misclassification could have occurred due to extended circulation of parasite antigens resulting in positive RDTs after successful treatment for a malaria episode. Although efforts were made during the trial to minimise this effect using blood smear confirmation of RDT-positive cases occurring within 2 weeks of a previous episode, another recent study showed the positive predictive value of an RDT relative to microscopy is still under 70% 5 weeks after an episode,³⁶ meaning that the incidence of uncomplicated malaria could be slightly overestimated.”

The authors make a good point of poor positive predictive value of RDTs compared to microscopy, although it is unclear if this should apply to the ~40-100 days window. The literature supports overestimation of malaria incidence by HRP2-based RDTs and while the current version of the manuscript does not state how many episodes were microscopically confirmed and how many were not, the deviation from model-predicted uncomplicated incidence remains mild. Being parsimonious in the number of parameters estimated is also good practice here, as the proposed PDMC model remains naïve to PK/PD data that were observed over the course of the trial.

Ultimately, the model-predicted incidence of hospitalized cases is spot-on, confirming the ability of the model to accurately replicate the outcome of interest (i.e. averting severe new episodes requiring hospitalization).

I am however not entirely convinced that the deviation, albeit small, can only be the result of misdiagnosis: should that be the case, a similar trend would be observed in the placebo arm as the number of episodes resembling clinical malaria with non-specific symptoms (e.g., fever) would be similar, assuming same pathogen pressure on the placebo and PDMC arms. Maybe the authors could consider slightly elaborating in the proposed discussion why this overestimation could be “diluted” in the placebo arm?

As a follow-up remark on the topic of “incidence”, L153-154 the authors state “This high incidence suggests persistent vulnerability of these children beyond the end of the PDMC intervention at week 14”. The authors should also consider spatial heterogeneity in exposure to malaria, in particular when comparing their placebo arm results to that from the general population.

2. The statement \The incidence of uncomplicated and hospitalised malaria ... per infectious bite decreased" isn't crystal clear to me. Do the authors mean that for a single individual, the more exposures that individual has, the risk of these outcomes decreases with each additional bite? And if so, is this because of the development of natural immunity?

We model this effect as a population average reduction in the probability of symptomatic malaria as EIR increases, rather than an individual increase in immunity with accumulated exposure, and we have now clarified the wording in the main text to reflect this. As the EIR increases, the incidence of symptomatic malaria (uncomplicated and hospitalised) has been observed to increase non-linearly, so that the incidence 'per bite' is lower.⁴ It may indeed result from the more rapid development of infection-blocking and anti-disease immunity in areas of high transmission. It may also result from density-dependent effects whereby incoming infections compete with one another and cannot all establish in the host simultaneously. Infectious bites close together in time may also simply create one symptomatic episode rather than several. We use one empirical function to capture these possible effects, taking a similar functional form to that used in previous analyses.^{5,6}

In the methods we edited the text as follows: "The *average* probability that an infectious bite leads to infection and symptoms are both known to decline with EIR in the general population *due to acquired immunity and density-dependent effects, e.g. competition between parasites*. We included this possibility by further scaling *total incidence in each location* by a similar functional form to that identified in previous analyses"

In the results we added: "The incidence of uncomplicated and hospitalised malaria increased with EIR, but the *average* risk of these outcomes per infectious bite decreased"

The authors rightly point in the direction that many factors shape the relationship between EIR and probability of successful infection upon an infectious mosquito bite. The explanation in the answer to Reviewer #1's comment is very clear, but the modifications in the main text are less (introducing the "average" word did not make the statement much clearer upon first read). Rather, I would suggest emphasizing that the proportion of infectious bites leading to a **successful** infection in humans decreases non-linearly as the EIR increases.

3. In the beginning of the Methods section, I recommend providing more details about the model structure. The authors describe that the model is a compartmental model, but it is not until the description of the Bayesian inference that the authors state \We coded the model in discrete time with a timestep of one day, with transition rates converted to daily probabilities." From the supplement, it appears that this model is deterministic, but it is worth noting in the Methods section of the main manuscript. Also, the authors state on p.23 that the full population model is a deterministic discrete-time model; I again suggest placing this statement towards the beginning of the model description.

*We have now moved this model description to the beginning of the methods: "We developed a *deterministic, discrete-time* compartmental model.."*

And a similar sentence at the beginning of the description of the full population model: “*This deterministic model was similarly coded in discrete time.*”

This modification perfectly addresses Reviewer #1’s suggestion.

In addition, I am wondering about the model implementation of several parameter that are indicated as “fixed” and that do indeed seem completely fixed in the GitHub repository, e.g. drug efficacy, duration of prophylaxis, etc. see https://github.com/lucyokell/pdmc_model/blob/14d87f093deb10ac97b692426da0cba262e63a06/pmc_model14g_cut8_correct_negbin.stan#L51). These are fixed to the value stated in the text (see for instance L450 for AL treatment and prophylactic effect). Did the author consider sampling these values from corresponding distributions (e.g. discretized Gamma with mean 13 days, possibly left-truncated to prevent low values) across their model iterations? This is still feasible without estimating these parameters, with the added benefit of introducing uncertainty around these point-estimates. This uncertainty would then be propagated to model outputs.

4. The authors fitted to daily incidence. What likelihood model (or heuristic objective function) was used? I could not find this stated in the main text nor the supplement. The authors used MCMC in the RStan software - is there a default objective function used here? This question is particularly important in the context of Figure 2, which indicates a much smaller uncertainty in the model output than the data suggest, indicating that the likelihood model is not chosen appropriately for the type of data.

Thank you for noticing this important omission – you are correct that we had not stated the likelihood model in the supplement and it was only available in the online code. We have now added a full description in the supplement (also copied below). In our original submission, we had modelled the daily incidence of uncomplicated or hospitalised malaria on a given day as Poisson distributed. The reviewer’s comment prompted us to re-examine the likelihood model in more detail. We have carried out further analysis and found that negative binomial likelihood gave a better fit to the data, as assessed by visual inspection of the posterior predictive checks (PPC) and the log-likelihood. We have now switched to using negative binomial likelihood instead and we have added a supplementary figure S8 showing the PPC of incidence in each site compared with the data values in each site as shown in Figure 2 (also copied below here). The figures show that the PPC and its uncertainty adequately encompass the majority of observed data. We thank the reviewer for identifying this issue which has improved the analysis.

The effect of changing the likelihood model on the posterior central estimates and distributions of key model parameters was not substantial (see comparison Table below) and therefore once propagated through to the rest of the results, did not produce any large change in the quantitative results in the manuscript (other than the posterior predictive intervals in Figure S8). The uncertainty in Figure 2 which the reviewer raises, remained similar. To clarify, the credible interval shown in this figure is the uncertainty around the mean prediction, rather than showing a posterior predictive interval, which we have now made clearer by revising the Figure 2 legend and referring the reader to the posterior

predictive checks in the supplement. We also now plot the posterior EIR on the x-axis of Figure 2 and added horizontal 95% CrI to better show the uncertainty in EIR (also copied here).

Changes in the manuscript:

[truncated for clarity]

I can only commend the author on going through new model estimates and goodness-of-fit comparisons in the light of Reviewer's #1 comment. The comparison table provided to compare Poisson to Negative-Binomial likelihood is reassuring with medians and 95%-CrI being largely similar. This table could actually make it into the Supplementary Information and referenced in the added paragraph describing the PDMC likelihood.

This paragraph, however, was extremely hard to read for multiple reasons and may contain errors. I apologize in advance to the authors if the presumed errors (point b. below) are the result of my misreading of indexes and multiplications (point a. below):

- a. First, the equations for W_q show a non-obvious mixture of brackets usage, some for indexing time t_d , some denoting a multiplication. This comment also applies for other equations (e.g., p.4 and p.8-11 in the Supplementary). Introducing proper multiplication signs where they are due would help the reader (or switching to a subscript notation for time?).
- b. Second, the equations for $W_1(t_d)$ and $W_3(t_d)$ are identical, and so are those for $W_2(t_d)$ and $W_4(t_d)$. Considering W_1 and W_3 refer to uncomplicated and hospitalized malaria infections in the placebo group, shouldn't they respectively implicate θ and $1-\theta$, respectively? The exact same consideration applies for W_2 and W_4 . There is good indication that this is indeed an error, as in the code implementation (https://github.com/lucyokell/pdmc_model/blob/14d87f093deb10ac97b692426da0c_ba262e63a06/pmc_model14g_cut8_correct_negbin.stan#L16) the authors define the right last term of the equation as complement probabilities.
- c. Third, the notation for Θ are not consistent between the equations (θ and θ_1) and Table S1 (θ_1 and θ_2). In the PDMC groups, θ_2 should rather be used for consistency.
- d. The notation for incidence Y is also inconsistent between the W equations (subscript d , indicating a reference to time? E.g. $y_d(t_d-1)$) and the likelihood equation (subscript q , denoting event).

This section, while extremely useful and a good model summary, would benefit from proof-reading and harmonization of notations. I would appreciate this section be reviewed again by the authors and would be happy to be proven wrong in my interpretation of the equations.

Other comments

The revised Figure 2, and part of the answer to point 4 just above, call for clarification / harmonization of values reported throughout the manuscript. I was surprised to read from the authors that the blue band in Figure 2A and 2B corresponds to a 95% CrI around the *mean*. Why not report medians and posterior CrI, as is usually done in most Bayesian frameworks? This would likely have individual data points (red) fall within the prediction CrI.

Furthermore, the manuscript sometimes mentions predicted “means”. Again, are these actually means, or medians (L233, L254, Figure S4 legend)? If they are indeed means, why are these reported instead of medians?

Overall, except for point 4 which requires confirmation from the author, I believe these comments have been properly addressed and while minor edits would be appreciated, the main findings are supported by the analyses.

Reviewer #2

Reviewer #2 did not make any comments on computational topics, but rather suggested two key areas of discussions to be added, around care-seeking behavior and related uncertainty.

There are no major flaws in the data analysis which i have noted, and the work is clearly written and well edited. One area where the authors might consider additional text/consideration is to highlight some areas of major uncertainty in the predictions of impact where either no data exists or the data that exists is of limited / poor quality. These are especially in the areas of hospital level care seeking among children with SMA, and in the adherence of these children to PDMC regimens in real world settings. Both of these parameters are unknown/incredibly uncertain and both are dramatically important to the estimation of population level effectiveness of these strategies.

We agree these are both extremely important uncertainties and we have added further text to the discussion to emphasise and address these as follows (underlined):

“Our results on the number of cases averted per child given PDMC and the number needed to treat to avert uncomplicated and hospitalised episodes were derived from clinical trial results and are relatively robust to model assumptions. However, the total demand for PDMC and the total number of cases and deaths that could be prevented are uncertain, given that PDMC is a hospital-based intervention and there is a lack of data on the probability of accessing hospital care. Access to hospitals is likely to vary greatly between and within countries.²⁸ A recent study tracking children with suspected severe malaria in the community in Uganda, Nigeria, and DRC found that only 41-65% access hospital care even after referral by a community health worker.²⁵ There are similar findings in LMIC settings relating to other diseases; for example in Lusaka, Zambia, only 32% of fatal respiratory syncytial virus cases in infants sought care in hospital before death. In our analysis we varied the percentage who access hospital care from 30-70% to allow for this uncertainty, however it is possible that in some countries, access is lower or higher, meaning that we over- or underestimate both demand and impact of PDMC.”

We added a further paragraph to the discussion to address the reviewer’s comment about potential adherence to PDMC during implementation:

“Our results are also sensitive to the assumed level of adherence to PDMC during routine implementation, which we based on an implementation trial that simply provided all treatments at discharge without further reminders.²⁴ 77-89% adherence was observed (adherence varied across the 1st-3rd treatments). If lower adherence was achieved during

routine implementation, we could over-estimate impact of PDMC. However, high adherence has generally been observed in other malaria chemoprevention interventions. For example routine seasonal malaria chemoprevention programmes in the Sahel region report adherence of 87-99%.³² “

The authors have addressed that point extensively, by providing care-seeking behavior available for SRV infections, affecting newborns and infants (a fraction of the population in the interest with respect to the present modelling exercise). The extremely large range of values explored backs up the sensitivity analysis mentioned in the legend for Figure 3, and with results in Figure S9. The legend for Figure 3 legend should state again the values for clarity (L245) or at least refer to Figure S9 right away and not at the end of the legend.

Lastly the authors might also consider including information on the relationship of the deaths prevented predicted in their models to the burden of malaria mortality in these areas of SSA. It seems clear but is not explicitly stated that this intervention will never have a major population burden impact, however, it seems to be incredibly effective way to target resources to a small but extremely vulnerable population and as such might be an incredibly efficient use of resources (where the health system is capable of delivering this with good uptake and adherence).

We agree with the reviewer's assessment on the population impact and mortality burden. We have added the following to the discussion to acknowledge this.

“Given that the vulnerable group targeted by PDMC is small, we estimate PDMC would prevent only a fraction of total population SMA cases (5-9% in high transmission areas) and a smaller fraction of all malaria deaths (WHO estimates ~427,200 malaria deaths in under fives in 2019). This highlights the importance of implementing PDMC alongside key preventive interventions across the whole community”

This comment addresses the reviewer's question adequately.

Point by point response to the reviewer.

We have labelled the comments as follows:

Reviewer comments in blue.

New author responses in black indented with >>>

Quotes from the previous round of reviews and our previous response in shaded boxes.

Review of the manuscript “Projected health impact of post-discharge malaria chemoprevention among children with severe malarial anaemia in Africa” by Okell *et al.*

Reviewer: Dr. Thomas Obadia

In their manuscript, Okell *et al.* estimate the impact of post-discharge malaria chemoprevention (PDMC) after severe malarial anaemia (SAM). The key findings highlight the substantial benefits of PDMC with low numbers of children treated required to reduce the number of hospital readmissions and subsequent deaths. While not dwelling into actual cost effectiveness analyses, it is undeniable that these results will be useful to policymakers to design and optimize public health control measures.

The manuscript models the outcomes of interest in a two-step process, with both steps relying on deterministic, compartmental models in a Bayesian setting. As a first step, the authors model the PDMC process in a longitudinal framework, with input being transmission (entomological infectious rate, EIR) at the locations from a multicenter clinical trial (PDMC vs. placebo). The outputs of interest are uncomplicated new malaria episodes and hospital readmissions. The model was able to accurately reproduce the trend between EIR and outcomes of interest. All 9 centers from the trial were used in that PDMC model while 3 additional trials (in Uganda and Malawi, conducted by co-authors of the present manuscript) were used as validation to assess model ability to predict incidence. In a second step, the authors embedded their PDMC model into a larger model of SMA in sub-Saharan Africa. This second model is not extensively described in the main text, but rather in the Supplementary Material: it essentially uses the same Bayesian framework and children, initially from the general population (G), can develop SMA and either go through hospital admission (H) and associated pathways (artemeter-lumefantrine prophylaxis, followed or not by PDMC as modeled before) or have their disease handled in the community only (Dc). Any SMA episode, regardless of its pathway (hospital or community) is followed by a susceptible period at high risk and another one at lower risk (subscript nHR). Recurrent SMA is defined as any transition from high risks compartment to a new SMA episode. Children may also not experience SMA in the following 6 months and they then return to the G state.

The key findings are well supported by the model and data, the scientific approach seems sound and the author clearly discuss the limitations of their modelling approach in the discussion, in particular with respect to generalization of some results due to uncertainty in key parameters like treatment-seeking behavior and adherence. The source code for the model has also been uploaded to a GitHub repository: this is an excellent initiative and while the repository would benefit from a README and short description to ease reproducibility, it is still rare enough of an initiative to be highlighted.

My review will build upon the comments from Reviewer #1 and Reviewer #2 and subsequent answers from the authors, as requested by the handling Editor. In the following,

the original reviewer comments and answers will be in separate boxes, and my comments will follow.

I am overall impressed with the amount of work and the writing of this manuscript, and I would like to see it published in Nature Communications as it will benefit its readership. I do however still have concerns that need to be addressed before publication can be recommended.

Reviewer #1

The only 2 minor comments have been properly addressed.

(Previous reviewer comment) 1. Fig 1C - what is the explanation for the discrepancy between the model fits and the data during the first 100 days post discharge?

(Previous author response) Thank you for this question. Figure 1C shows the cumulative daily number of uncomplicated cases in the placebo and PDMC trial arms, comparing data and model fits. The placebo arm model fit is very close to the placebo arm data values. However in the first 100 days, the PDMC trial arm has a slightly higher number of episodes in the data than are predicted by the model fit, as the reviewer comments. The first 100 days are approximately the 14-week (98 days) period during which participants were protected from malaria. This includes 2 weeks of post-treatment prophylaxis following artemether-lumefantrine at discharge, and then an additional 12 weeks by providing 3 courses of PDMC with dihydroartemisinin piperaquine (DP) at the end of the 2nd, 6th and 10th week post-discharge. Each course of DP provides approximately 4 weeks of post-treatment prophylaxis. The reason for the imperfect fit is that we used fixed parameters to describe the duration and strength of protection provided by the PDMC drug, dihydroartemisinin-piperaquine (DP), from a previous analysis (1 – see also methods section). We could have chosen to re-estimate the DP drug protection parameters using the PDMC trial data, which would have provided a better fit during these first 100 days. However on balance we felt this could qualify as overfitting of the model, a concern also raised by reviewer 3. We considered the previous analysis which estimated the DP parameters to be robust since it was based on a large dataset containing 8 trial sites, and was able to capture trends well across these varied sites with different transmission. Overall the fit to the current data was close enough that we did not feel justified re-fitting the drug parameters, particularly since our main outcome of interest was prevention of hospitalised episodes, where the model did fit the data well (Figure 1B).

An additional reason for retaining drug parameters from the previous analysis was the diagnostics used for uncomplicated malaria cases during the PDMC trial – some patients were diagnosed with HRP-2 rapid diagnostic tests while others were diagnosed by microscopy. Rapid diagnostic tests are known to have poor predictive value after a recent malaria episode, with individuals remaining positive for around 2 weeks after successful clearance of parasites following treatment of malaria (due to residual antigens rather than live parasites). Similar issues have been encountered in seasonal malaria chemoprevention trials (see reference 2 – their Supplementary Table 2).

*In the PDMC cohort, which was highly exposed to malaria, there may have therefore been some false positives where children had fever from another cause and tested positive by HRP2-based RDT due to a recent rather than current episode of malaria. Although efforts were made during the trial to minimise this effect by confirming infections with microscopy if occurring within 2 weeks of a previous episode, this does not perfectly solve the issue since some individuals remain RDT-positive beyond 2 weeks. This could have slightly overestimated the incidence of uncomplicated malaria. There is insufficient data to estimate this effect in this study, since asymptomatic infection was not tracked during the trial. However we have added an additional explanation of this in the discussion section as follows:
“Some further misclassification could have occurred due to extended circulation of parasite antigens resulting in positive RDTs after successful treatment for a malaria episode. Although efforts were made during the trial to minimise this effect using blood smear confirmation of RDT-positive cases occurring within 2 weeks of a previous episode, another recent study showed the positive predictive value of an RDT relative to microscopy is still under 70% 5 weeks after an episode,³⁶ meaning that the incidence of uncomplicated malaria could be slightly overestimated.”*

The authors make a good point of poor positive predictive value of RDTs compared to microscopy, although it is unclear if this should apply to the ~40-100 days window. The literature supports overestimation of malaria incidence by HRP2-based RDTs and while the current version of the manuscript does not state how many episodes were microscopically confirmed and how many were not, the deviation from model-predicted uncomplicated incidence remains mild. Being parsimonious in the number of parameters estimated is also good practice here, as the proposed PDMC model remains naive to PK/PD data that were observed over the course of the trial.

Ultimately, the model-predicted incidence of hospitalized cases is spot-on, confirming the ability of the model to accurately replicate the outcome of interest (i.e. averting severe new episodes requiring hospitalization).

I am however not entirely convinced that the deviation, albeit small, can only be the result of misdiagnosis: should that be the case, a similar trend would be observed in the placebo arm as the number of episodes resembling clinical malaria with non-specific symptoms (e.g., fever) would be similar, assuming same pathogen pressure on the placebo and PDMC arms. Maybe the authors could consider slightly elaborating in the proposed discussion why this overestimation could be “diluted” in the placebo arm?

>>>> Thank you for this good point raised. We consider the 40-90 days window potentially relevant for false RDT positives given the regular use of chemoprevention with DP in the PDMC arm during this time (at days 0,30,60). This may result in larger numbers of recently cleared infections in this group. We now add this possible explanation for why this effect would be lower in the placebo arm than the PDMC arm, to the discussion as follows (new text in **bold**):

“Some further misclassification could have occurred due to extended circulation of parasite antigens resulting in positive RDTs after successful treatment for a malaria episode. Although efforts were made during the trial to minimise this effect using blood smear confirmation of RDT-positive cases occurring within 2 weeks of a previous episode, another recent study showed the positive predictive

value of an RDT relative to microscopy is still under 70% 5 weeks after an episode,³⁶ meaning that the incidence of uncomplicated malaria could be slightly overestimated. **This effect may be stronger in the PDMC arm given their additional intake of drugs as part of the intervention, which would result in a higher prevalence of recently cleared infection. This may explain the slight deviation of the model from the data in in the PDMC arm during the intervention period (Figure 1).**”

As a follow-up remark on the topic of “incidence”, L153-154 the authors state “This high incidence suggests persistent vulnerability of these children beyond the end of the PDMC intervention at week 14”. The authors should also consider spatial heterogeneity in exposure to malaria, in particular when comparing their placebo arm results to that from the general population.

>>> Thank you, this is a good point and we agree may well be a contributing factor. We have now added in the following on spatial heterogeneity as follows (new text in **bold**):

“This high incidence suggests persistent vulnerability of these children beyond the end of the PDMC intervention at week 14. The incidence of uncomplicated malaria was 1.2-2.5 times higher than expected in the general population of the same age in weeks 3-14. The total incidence of symptomatic malaria episodes (both uncomplicated and hospitalised) was more than the expected incidence of infectious bites in 0-5 year olds in four trial sites, suggestive of higher than average exposure to mosquitoes (**e.g. due to spatial heterogeneity in transmission, lack of protective measures such as bed nets, etc**) (Figure S6).”

2. **(Previous reviewer comment)** *The statement “The incidence of uncomplicated and hospitalised malaria ... per infectious bite decreased” isn't crystal clear to me. Do the authors mean that for a single individual, the more exposures that individual has, the risk of these outcomes decreases with each additional bite? And if so, is this because of the development of natural immunity?*

(Previous author response) *We model this effect as a population average reduction in the probability of symptomatic malaria as EIR increases, rather than an individual increase in immunity with accumulated exposure, and we have now clarified the wording in the main text to reflect this. As the EIR increases, the incidence of symptomatic malaria (uncomplicated and hospitalised) has been observed to increase non-linearly, so that the incidence ‘per bite’ is lower.⁴ It may indeed result from the more rapid development of infection-blocking and anti-disease immunity in areas of high transmission. It may also result from density-dependent effects whereby incoming infections compete with one another and cannot all establish in the host simultaneously. Infectious bites close together in time may also simply create one symptomatic episode rather than several. We use one empirical function to capture these possible effects, taking a similar functional form to that used in previous analyses.^{5,6} In the methods we edited the text as follows:*

*“The **average** probability that an infectious bite leads to infection and symptoms are both known to decline with EIR in the general population **due to acquired immunity and density-dependent effects, e.g. competition between parasites**. We included this possibility by further scaling **total incidence in each location** by a similar functional form to that identified in previous analyses⁷” In the results we added: “The incidence of uncomplicated and hospitalised malaria increased with EIR, but the **average** risk of these outcomes per infectious bite decreased”*

The authors rightly point in the direction that many factors shape the relationship between EIR and probability of successful infection upon an infectious mosquito bite. The explanation in the answer to Reviewer #1's comment is very clear, but the modifications in the main text are less (introducing the "average" word did not make the statement much clearer upon first read). Rather, I would suggest emphasizing that the proportion of infectious bites leading to a **successful** infection in humans decreases non-linearly as the EIR increases.

>>> Thanks for this suggestion, we have incorporated this in the text as follows (new text in **bold**):

"The average probability that an infectious bite leads to **successful** infection and **then to** symptoms are both known to decline with increasing transmission intensity in the general population in any given setting due to acquired immunity and density-dependent effects, e.g. competition between parasites."

3. (Previous reviewer comment) In the beginning of the Methods section, I recommend providing more details about the model structure. The authors describe that the model is a compartmental model, but it is not until the description of the Bayesian inference that the authors state \We coded the model in discrete time with a timestep of one day, with transition rates converted to daily probabilities." From the supplement, it appears that this model is deterministic, but it is worth noting in the Methods section of the main manuscript. Also, the authors state on p.23 that the full population model is a deterministic discrete-time model; I again suggest placing this statement towards the beginning of the model description.

(Previous author response) We have now moved this model description to the beginning of the methods: "We developed a deterministic, discrete-time compartmental model.."

And a similar sentence at the beginning of the description of the full population model: *"This deterministic model was similarly coded in discrete time."*

This modification perfectly addresses Reviewer #1's suggestion.

In addition, I am wondering about the model implementation of several parameter that are indicated as "fixed" and that do indeed seem completely fixed in the GitHub repository, e.g. drug efficacy, duration of prophylaxis, etc. see

https://github.com/lucyokell/pdmc_model/blob/14d87f093deb10ac97b692426da0cba262e63a06/pmc_model14g_cut8_correct_negbin.stan#L51). These are fixed to the value stated in the text (see for instance L450 for AL treatment and prophylactic effect). Did the author consider sampling these values from corresponding distributions (e.g. discretized Gamma with mean 13 days, possibly left-truncated to prevent low values) across their model iterations?

This is still feasible without estimating these parameters, with the added benefit of introducing uncertainty around these point-estimates. This uncertainty would then be propagated to model outputs.

>>>> Thank you for raising this query. The reviewer is correct that there are some fixed parameters, while others are implemented as distributions or are estimated during MCMC. The reasons for these choices are in part computational and in part dependent on how important the parameter is to this analysis or how much is already known about its value.

We opted to fix the drug efficacy parameters since they had been previously well characterised in a systematic review of a large number of studies (at least 65 per drug)¹, so that the uncertainty is extremely small relative to other larger uncertainties in the analysis.

The drug durations of prophylaxis were straightforward to implement as distributions of times within the population in this analysis (as the reviewer favours) when the whole cohort is treated **at the same time**. So we implemented AL prophylaxis provided by treatment at discharge as a gamma distribution of times, since all children are treated on their first day of follow up (day of discharge). I.e. we move a fixed proportion of the group into the susceptible state each day according to the probability of the gamma distribution on that day. DP treatment is similarly given at 3 specific times during follow up which are the same for each child. DP prophylaxis is therefore modelled as a Weibull survival function which provides a probability of protection, so that some proportion of treated individuals have longer durations of protection than others. This probability can be applied to the whole treated cohort in the PDMC arm on a given day of follow up. The drug functions are parameterised from our previous analysis.²

However, AL treatment was also given during the trial follow up for episodes of uncomplicated malaria, and we allowed that this would also provide a period of prophylaxis. Since these treatments occur on different follow up days for different children, it is more difficult to implement a distribution of times here. We could have chosen to use an Erlang distribution i.e. multiple compartments to approximate the AL duration of prophylaxis, but this would have slowed MCMC fitting and the Erlang also has the constraints of an integer scale parameter. Therefore we implemented a fixed duration of protection arising from treatment of new cases with AL during follow up.

In terms of overall results, uncertainty over AL prophylactic duration is a very small part of the overall uncertainty in total population impact of PDMC. We show below the gamma distribution of the duration of AL protection estimated in our previous work, with relatively small variance in the distribution. By contrast, there are very large uncertainties in the population model around the proportion of children who access hospital care when severely ill (our reasonable range is 30-70%) which make a very large difference to the outcome (e.g. Figure S10). For this reason we felt it was reasonable to fix the drug efficacy and duration parameters.

¹ 'World Malaria Report 2020. Geneva: World Health Organization.'

² Okell et al., 'Contrasting Benefits of Different Artemisinin Combination Therapies as First-Line Malaria Treatments Using Model-Based Cost-Effectiveness Analysis'.

Figure: duration of AL protection: gamma distribution estimated from data in our previous paper.³

Without re-fitting the parameters, there is not a standard way as far as we are aware to sample different values for the AL prophylaxis duration (as the reviewer may be suggesting) within a standard programming language for statistical inference, such as Stan (which we used for our analysis). In Stan, random number generation is not allowed within the transformed parameters nor the model blocks of the code, presumably because this would make the model convergence difficult.

4. (Previous reviewer comment) *The authors fitted to daily incidence. What likelihood model (or heuristic objective function) was used? I could not find this stated in the main text nor the supplement. The authors used MCMC in the RStan software - is there a default objective function used here? This question is particularly important in the context of Figure 2, which indicates a much smaller uncertainty in the model output than the data suggest, indicating that the likelihood model is not chosen appropriately for the type of data.*

(Previous author response) *Thank you for noticing this important omission – you are correct that we had not stated the likelihood model in the supplement and it was only available in the online code. We have now added a full description in the supplement (also copied below). In our original submission, we had modelled the daily incidence of uncomplicated or hospitalised malaria on a given day as Poisson distributed. The reviewer’s comment prompted us to re-examine the likelihood model in more detail. We have carried out further analysis and found that negative binomial likelihood gave a better fit to the data, as assessed by visual inspection of the posterior predictive checks (PPC) and the log-likelihood. We have now switched to using negative binomial likelihood instead and we have added a supplementary figure S8 showing the PPC of incidence in each site compared with the data values in each site as shown in Figure 2 (also copied below here). The figures show that the PPC and its uncertainty adequately encompass the majority of observed data. We thank the reviewer for identifying this issue which has improved the analysis.*

³ Okell et al.

The effect of changing the likelihood model on the posterior central estimates and distributions of key model parameters was not substantial (see comparison Table below) and therefore once propagated through to the rest of the results, did not produce any large change in the quantitative results in the manuscript (other than the posterior predictive intervals in Figure S8). The uncertainty in Figure 2 which the reviewer raises, remained similar. To clarify, the credible interval shown in this figure is the uncertainty around the mean prediction, rather than showing a posterior predictive interval, which we have now made clearer by revising the Figure 2 legend and referring the reader to the posterior predictive checks in the supplement. We also now plot the posterior EIR on the x-axis of Figure 2 and added horizontal 95% CrI to better show the uncertainty in EIR (also copied here).

*Changes in the manuscript:
[truncated for clarity]*

I can only commend the author on going through new model estimates and goodness-of-fit comparisons in the light of Reviewer's #1 comment. The comparison table provided to compare Poisson to Negative-Binomial likelihood is reassuring with medians and 95%-CrI being largely similar. This table could actually make it into the Supplementary Information and referenced in the added paragraph describing the PDMC likelihood.

>>> We have added this sentence to the supplementary information paragraph on the likelihood: "Parameter estimates and posterior distributions were very similar if a Poisson versus a negative binomial distribution were assumed, but the maximum likelihood was higher for the negative binomial fit."

This paragraph, however, was extremely hard to read for multiple reasons and may contain errors. I apologize in advance to the authors if the presumed errors (point b. below) are the result of my misreading of indexes and multiplications (point a. below):

a. First, the equations for W_q show a non-obvious mixture of brackets usage, some for indexing time t_d , some denoting a multiplication. This comment also applies for other equations (e.g., p.4 and p.8-11 in the Supplementary). Introducing proper multiplication signs where they are due would help the reader (or switching to a subscript notation for time?).

>>>> Thank you for pointing this out, and we agree the mixture of brackets made the equations unclear. We have edited these equations and opted to use round brackets only for indexing time, and to use square brackets in other cases, as the simplest and most standard notation option. This is now harmonised through the supplementary material.

b. Second, the equations for $W_1(t_d)$ and $W_3(t_d)$ are identical, and so are those for $W_2(t_d)$ and $W_4(t_d)$. Considering W_1 and W_3 refer to uncomplicated and hospitalized malaria infections in the placebo group, shouldn't they respectively implicate θ and $1-\theta$, respectively? The exact same consideration applies for W_2 and W_4 . There is good indication that this is indeed an error, as in the code implementation (https://github.com/lucyokell/pdmc_model/blob/14d87f093deb10ac97b692426da0cba262e63a06/pmc_model14g_cut8_correct_negbin.stan#L16) the authors define the right last term of the equation as complement probabilities.

>>>> Many thanks for spotting this error on our part. Indeed θ indicates the probability of hospitalisation, and W3 and W4 are the uncomplicated cases which are not hospitalised. So the reviewer is correct we should have used $(1 - \theta)$ in these equations. This is now corrected.

c. Third, the notation for Thetas are not consistent between the equations (θ and θ_1) and Table S1 (θ_1 and θ_2). In the PDMC groups, θ_2 should rather be used for consistency.

>>>> Thank you for this good query and we agree it was unclear. This is a slight misunderstanding from the reviewer as the θ_2 is not used for the PDMC groups throughout the simulation. We have now added the following clarification underneath the equations: “where the probability of a symptomatic episode being more severe and requiring hospitalisation in the PDMC group, θ , is θ_2 when the child is protected by DP with probability of protection $p_{pDP} > 0.01$ and θ_1 when $p_{pDP} \leq 0.01$.”

d. The notation for incidence Y is also inconsistent between the W equations (subscript d , indicating a reference to time? E.g. $y_d(t_d-1)$) and the likelihood equation (subscript q , denoting event).

>>>> We have changed the notation here. The y_d denotes the daily probability of an episode and we agree it was therefore confusing to use capital Y to indicate a different quantity (number of events). We changed Y_q to X_q since we have not used the letter ‘x’ elsewhere in the notation.

This section, while extremely useful and a good model summary, would benefit from proofreading and harmonization of notations. I would appreciate this section be reviewed again by the authors and would be happy to be proven wrong in my interpretation of the equations.

>>>> We are grateful for a thorough review and have made all the suggested changes. We have also proofread this section again.

Other comments

The revised Figure 2, and part of the answer to point 4 just above, call for clarification / harmonization of values reported throughout the manuscript. I was surprised to read from the authors that the blue band in Figure 2A and 2B corresponds to a 95% CrI around the *mean*.

Why not report medians and posterior CrI, as is usually done in most Bayesian frameworks? This would likely have individual data points (red) fall within the prediction CrI.

Furthermore, the manuscript sometimes mentions predicted “means”. Again, are these actually means, or medians (L233, L254, Figure S4 legend)? If they are indeed means, why are these reported instead of medians?

Overall, except for point 4 which requires confirmation from the author, I believe these comments have been properly addressed and while minor edits would be appreciated, the main findings are supported by the analyses.

>>>> Thank you for spotting this error in terminology. We did indeed use and report the medians and CrI as is standard. We have now corrected the mention of the mean in Figure 2 and in the supplement. In our previous response we wished to distinguish between the standard posterior CrI (derived from uncertainty in the parameters), versus the posterior predictive interval (which additionally takes into account sampling variation in the data as well as the parameters). We plotted the posterior CrI in Figure 2, but cannot plot the

predictive interval in the model fit there due to the lack of data across all EIR values. However we show the site-specific posterior predictive intervals in the Supplement (Figure S8).

Reviewer #2

Reviewer #2 did not make any comments on computational topics, but rather suggested two key areas of discussions to be added, around care-seeking behavior and related uncertainty.

(Previous reviewer comment) *There are no major flaws in the data analysis which i have noted, and the work is clearly written and well edited. One area where the authors might consider additional text/consideration is to highlight some areas of major uncertainty in the predictions of impact where either no data exists or the data that exists is of limited / poor quality. These are especially in the areas of hospital level care seeking among children with SMA, and in the adherence of these children to PDMC regimens in real world settings. Both of these parameters are unknown/incredibly uncertain and both are dramatically important to the estimation of population level effectiveness of these strategies.*

(Previous author response) *We agree these are both extremely important uncertainties and we have added further text to the discussion to emphasise and address these as follows: "Our results on the number of cases averted per child given PDMC and the number needed to treat to avert uncomplicated and hospitalised episodes were derived from clinical trial results and are relatively robust to model assumptions. However, the total demand for PDMC and the total number of cases and deaths that could be prevented are uncertain, given that PDMC is a hospital-based intervention and there is a lack of data on the probability of accessing hospital care. Access to hospitals is likely to vary greatly between and within countries.²⁸ A recent study tracking children with suspected severe malaria in the community in Uganda, Nigeria, and DRC found that only 41-65% access hospital care even after referral by a community health worker.²⁵ There are similar findings in LMIC settings relating to other diseases; for example in Lusaka, Zambia, only 32% of fatal respiratory syncytial virus cases in infants sought care in hospital before death. In our analysis we varied the percentage who access hospital care from 30-70% to allow for this uncertainty, however it is possible that in some countries, access is lower or higher, meaning that we over- or underestimate both demand and impact of PDMC."*

We added a further paragraph to the discussion to address the reviewer's comment about potential adherence to PDMC during implementation:

"Our results are also sensitive to the assumed level of adherence to PDMC during routine implementation, which we based on an implementation trial that simply provided all treatments at discharge without further reminders.²⁴ 77-89% adherence was observed (adherence varied across the 1st-3rd treatments). If lower adherence was achieved during routine implementation, we could over-estimate impact of PDMC. However, high adherence has generally been observed in other malaria chemoprevention interventions. For example routine seasonal malaria chemoprevention programmes in the Sahel region report adherence of 87-99%.³² "

The authors have addressed that point extensively, by providing care-seeking behavior available for SRV infections, affecting newborns and infants (a fraction of the population in

the interest with respect to the present modelling exercise). The extremely large range of values explored backs up the sensitivity analysis mentioned in the legend for Figure 3, and with results in Figure S9. The legend for Figure 3 legend should state again the values for clarity (L245) or at least refer to Figure S9 right away and not at the end of the legend.

>>>> We have now added details of the sensitivity analysis here nearer the start of the legend for Figure 3: “The assumption in these results is that 50% of cases requiring hospitalisation access hospital care, but there is negligible change in these outputs when this percentage is varied from 30-70%.” Figure S9 is a sensitivity analysis of Figure 3D, so we reference it in the relevant part of the legend.

(Previous reviewer comment) *Lastly the authors might also consider including information on the relationship of the deaths prevented predicted in their models to the burden of malaria mortality in these areas of SSA. It seems clear but is not explicitly stated that this intervention will never have a major population burden impact, however, it seems to be an incredibly effective way to target resources to a small but extremely vulnerable population and as such might be an incredibly efficient use of resources (where the health system is capable of delivering this with good uptake and adherence).*

(Previous author reply) *We agree with the reviewer’s assessment on the population impact and mortality burden.*

We have added the following to the discussion to acknowledge this.

“Given that the vulnerable group targeted by PDMC is small, we estimate PDMC would prevent only a fraction of total population SMA cases (5-9% in high transmission areas) and a smaller fraction of all malaria deaths (WHO estimates ~427,200 malaria deaths in under fives in 2019). This highlights the importance of implementing PDMC alongside key preventive interventions across the whole community”

This comment addresses the reviewer’s question adequately.